# Glucocerebrosidase is imported into mitochondria and preserves complex I integrity and energy metabolism

Pascale Baden[1,2,3,10], Maria Jose Perez[1,2,3,10], Hariam Raji[1,2,3,11], Federico Bertoli[1,2,3,11], Stefanie Kalb[1,2,11], María Illescas [4], Fokion Spanos [1,2,3], Claudio Giuliano[1,2,5], Alessandra Maria Calogero [6], Marvin Oldrati [1,2,3], Hannah Hebestreit[1,2], Graziella Cappelletti [6], Kathrin Brockmann[1,2], Thomas Gasser [1,2], Anthony H. V. Schapira [3,7], Cristina Ugalde [4,8] & Michela Deleidi [1,2,3,9] ✉

Mutations in *GBA1*, the gene encoding the lysosomal enzyme β-glucocerebrosidase (GCase), which cause Gaucher's disease, are the most frequent genetic risk factor for Parkinson's disease (PD). Here, we employ global proteomic and single-cell genomic approaches in stable cell lines as well as induced pluripotent stem cell (iPSC)-derived neurons and midbrain organoids to dissect the mechanisms underlying GCase-related neurodegeneration. We demonstrate that GCase can be imported from the cytosol into the mitochondria via recognition of internal mitochondrial targeting sequence-like signals. In mitochondria, GCase promotes the maintenance of mitochondrial complex I (CI) integrity and function. Furthermore, GCase interacts with the mitochondrial quality control proteins HSP60 and LONP1. Disease-associated mutations impair CI stability and function and enhance the interaction with the mitochondrial quality control machinery. These findings reveal a mitochondrial role of GCase and suggest that defective CI activity and energy metabolism may drive the pathogenesis of GCase-linked neurodegeneration.

Mutations in the acid β-glucocerebrosidase (*GBA1*) gene cause the lysosomal storage disease Gaucher's disease (GD), which is an autosomal recessive condition that can present with both systemic and neurological symptoms. Three clinical GD subtypes have been identified: nonneuropathic (type I), acute neuropathic (type II), and chronic neuropathic (type III). *GBA1* mutations have been classified as mild or severe according to their association with GD type I or II and III. After the initial reports of an increased incidence of Parkinson's disease (PD) in patients affected by GD type I and in their family members, *GBA1* mutations have been identified as the most frequent genetic risk factor for PD[1,2]. To date, more than 350 pathogenic *GBA1* mutations have been linked to GD, some of which are also linked to PD. Mutations

[1]German Center for Neurodegenerative Diseases (DZNE), Tübingen, Germany. [2]Department of Neurodegenerative Diseases, Center of Neurology, Hertie Institute for Clinical Brain Research, University of Tübingen, Tübingen, Germany. [3]Aligning Science Across Parkinson's (ASAP) Collaborative Research Network, Chevy Chase, MD 20815, USA. [4]Instituto de Investigación Hospital 12 de Octubre (i + 12), Madrid 28041, Spain. [5]Unit of Cellular and Molecular Neurobiology, IRCCS Mondino Foundation, 27100 Pavia, Italy. [6]Department of Biosciences, Center of Excellence on Neurodegenerative Diseases, Università degli Studi di Milano, Milan, Italy. [7]Department of Clinical and Movement Neurosciences, University College London Queen Square Institute of Neurology, Royal Free Campus, London NW3 2PF, UK. [8]Centro de Investigación Biomédica en Red de Enfermedades Raras (CIBERER), U723 Madrid, Spain. [9]Present address: Institut Imagine, INSERM UMR1163 Paris Cite' University, 24 boulevard du Montparnasse, 75015 Paris, France. [10]These authors contributed equally: Pascale Baden, Maria Jose Perez. [11]These authors jointly supervised this work: Hariam Raji, Federico Bertoli, Stefanie Kalb. ✉e-mail: michela.deleidi@institutimagine.org

associated with the most severe neuronopathic forms of GD are linked to a higher risk of developing PD[3]. Among *GBA1* mutations, four missense variants (p.E326K, p.T369M, p.N370S, and p.L444P) account for ~87% of PD cases[4]. Interestingly, *GBA1* variants commonly observed in PD patients, such as the p.E326K variant, are not associated with GD[5,6]. The p.E326K variant displays a limited impact on GCase activity and a lower penetrance compared to the severe mutation p.L444P[7,8]. PD patients carrying E326K mutations have milder motor complications and a higher risk of cognitive decline[8,9]. Despite the link between *GBA1* and brain disease, the mechanisms underlying neurodegeneration and disease severity in mutation carriers are still elusive. *GBA1* encodes glucocerebrosidase (GCase), a lysosomal enzyme that catalyzes the hydrolysis of glucosylceramide into glucose and ceramide. *GBA1* effects may arise from several pathways, including lysosomal dysfunction, sphingolipid dyshomeostasis, α-synuclein (A-SYN) aggregation, and defects in autophagy and protein trafficking[10].

Here, we dissected *GBA1*-related neurodegeneration by investigating GCase-interacting proteins via a global proteomic approach. Furthermore, we explored how distinct disease-causing mutations alter the profile of GCase interactions. Using molecular and biochemical approaches, we identified a function of GCase as a mitochondrial protein with a role in maintaining respiratory chain complex I (CI) integrity and cellular energy homeostasis in cell lines as well as a patient-induced pluripotent stem cell (iPSC)-derived dopaminergic (DA) neurons and midbrain organoids. Furthermore, we provide a mechanism for mitochondrial LONP1 protease in the folding and degradation of mitochondrial GCase. Importantly, by combining brain organoids with single-cell RNA sequencing (RNA-Seq), we provide evidence for the role of GCase in the disruption of neuronal CI function and mitochondrial energy metabolism.

## Results

### Analysis of the GCase interactome in inducible T-Rex HEK cell lines reveals novel mitochondrial interactors

To obtain insight into the molecular mechanisms that link GCase to brain disease, we set out to identify the interactome of wild-type (WT) and mutant GCase using quantitative proteomic analysis. As most *GBA1* mutations lead to a misfolded enzyme that is rapidly degraded via ER-associated degradation by the proteasome[11,12], we generated stable Flp-In™T-REx™-HEK 293 cell lines (henceforth, T-Rex HEK cells) expressing WT or mutant GCase (E326K or L444P) as a V5-FLAG-tagged protein using a tetracycline-inducible system (Fig. 1A). This model provided a sufficient level of expression of both WT and mutant GCase (Fig. 1B–D). GCase protein levels and activity were significantly decreased in E326K- and L444P- compared to WT-GCase T-Rex HEK cells, and the degree of reduction was in accordance with the severity of the mutation (Fig. 1C, D). In line with these data, lysosomal GCase localization was decreased in mutant GCase cells (Supplementary Fig. 1A). Next, we performed FLAG immunoprecipitation followed by isobaric labeling (tandem mass tag, TMT) and MS analysis. T-Rex HEK cells expressing the empty vector were used as a control. GCase interactors were defined as proteins that were quantified by ≥2 unique peptides and enriched 1.5-fold over the control (Supplementary Dataset 1). First, we investigated the interactome of WT-GCase. The compiled list contained known GCase interactors, such as LIMP2, calnexin, calreticulin, and DNAJB11. Analysis of the 100 top hits of the WT-GCase interactome showed that proteins from different compartments, including the ER, Golgi, lysosomes, and vesicles, interact with GCase (Supplementary Fig. 1B). Interestingly, 19 out of the top 100 binding partners of WT-GCase were proteins associated with mitochondria (Supplementary Fig. 1B and Supplementary Dataset 1). The enrichment analysis of the biological processes (BP) revealed several mitochondrial pathways converging at energy metabolism (Fig. 1E). Interestingly, the analysis of the only interactome study of endogenous WT-GCase, which was

performed using SILAC proteomics in HeLa cells[13], revealed that 8% of endogenous GCase-interacting proteins were mitochondrial proteins (Supplementary Fig. 1C). Using GCase-FLAG immunoprecipitation followed by western blotting, we validated the interaction with cytosolic chaperones involved in protein folding prior to mitochondrial import (HSC70) as well as with mitochondrial outer membrane (TOM70), inner membrane (TIM23 and ATP5B), and mitochondrial matrix (HSP60 and LONP1) proteins (Fig. 1F and Supplementary Fig. 1D). Furthermore, we confirmed the specific interactions of WT-GCase with selected proteins from the cytosol, ER, and mitochondria (Supplementary Fig. 1E).

### Interactome and whole proteome analysis reveals the enrichment of mitochondrial proteins in L444P- and E326K-GCase mutants

Next, we compared the TMT-based quantitative interactomes of WT-, E326K-, and L444P-GCase in T-Rex HEK cells. Interestingly, among the dysregulated interactors of mutant GCase, we identified an enriched interaction with proteins involved in mitochondrial protein quality control, including LONP1, as well as a decreased interaction with the mitochondrial CI assembly factor TIMMDC1 (Fig. 1G and Supplementary Dataset 1). In addition, we performed a proteome-wide analysis of TMT-labeled whole-cell lysates from WT-, E326K-, and L444P-GCase T-Rex HEK cells. The analysis of the differentially expressed proteins (DEPs) revealed an increased abundance of mitochondrial proteins in WT-, E326K-, and L444P-GCase T-Rex HEK cells compared to the control (15, 25, and 20%, respectively) (Supplementary Fig. 2A, B and Supplementary Dataset 2). These included representative CI subunits from the matrix (NDUFA7, NDUFA12, NDUFS4, NDUFS5, and NDUFA10) and membrane arms (NDUFC2 and NDUFB8). In agreement, gene ontology (GO) analysis of DEPs revealed dysregulation of proteins linked to mitochondrial energy metabolism, especially to CI and the mitochondrial respirasome, in the E326K- and L444P-GCase proteomes (Supplementary Fig. 2C).

### GCase localizes to mitochondria

The TMT-based interactome dataset supports the hypothesis that both WT and mutant GCase can be imported into mitochondria. To confirm the mitochondrial localization of GCase, we performed subcellular fractionation of WT-, E326K-, and L444P-GCase T-Rex HEK cells, and we subjected isolated mitochondria to digitonin permeabilization and proteinase K digestion followed by western blotting (Fig. 2A and Supplementary Fig. 3A). Possible contamination of the mitochondrial fraction with lysosomal proteins was prevented by digitonin lysis and proteinase K digestion. The lysosomal proteins hexosaminidase B (HexB) and LAMP1 were successfully removed by this treatment, whereas GCase and the mitochondrial matrix protein LONP1 were still present in this fraction (Fig. 2A, and Supplementary Fig. 3A). To confirm the mitochondrial localization of GCase, T-Rex HEK cells overexpressing WT or mutant GCase were stained for TOM20 and GCase followed by expansion microscopy (ExM), which allows super-resolution imaging on a conventional fluorescence microscope[14] (Supplementary Fig. 3B).

### GCase contains internal MTS-like signals

As GCase does not have a conventional N-terminal mitochondrial targeting signal (MTS), we employed the TargetP predictive score consecutively for each residue to identify potential internal MTS-like signals (iMTS-ls)[15]. TargetP analysis revealed the presence of two iMTS-ls sites with a score >0.6 (Fig. 2B). To validate the potential role of these iMTS-ls in GCase mitochondrial import, we employed the split-GFP system[16], where the first ten β-strands of GFP (GFP$_{1-10}$) were targeted to mitochondria through linkage with a mitochondria-targeting sequence (MTS-GFP$_{1-10}$). MTS-GFP$_{1-10}$ was integrated into T-Rex HEK cells under a doxycycline-inducible promoter (Fig. 2C)

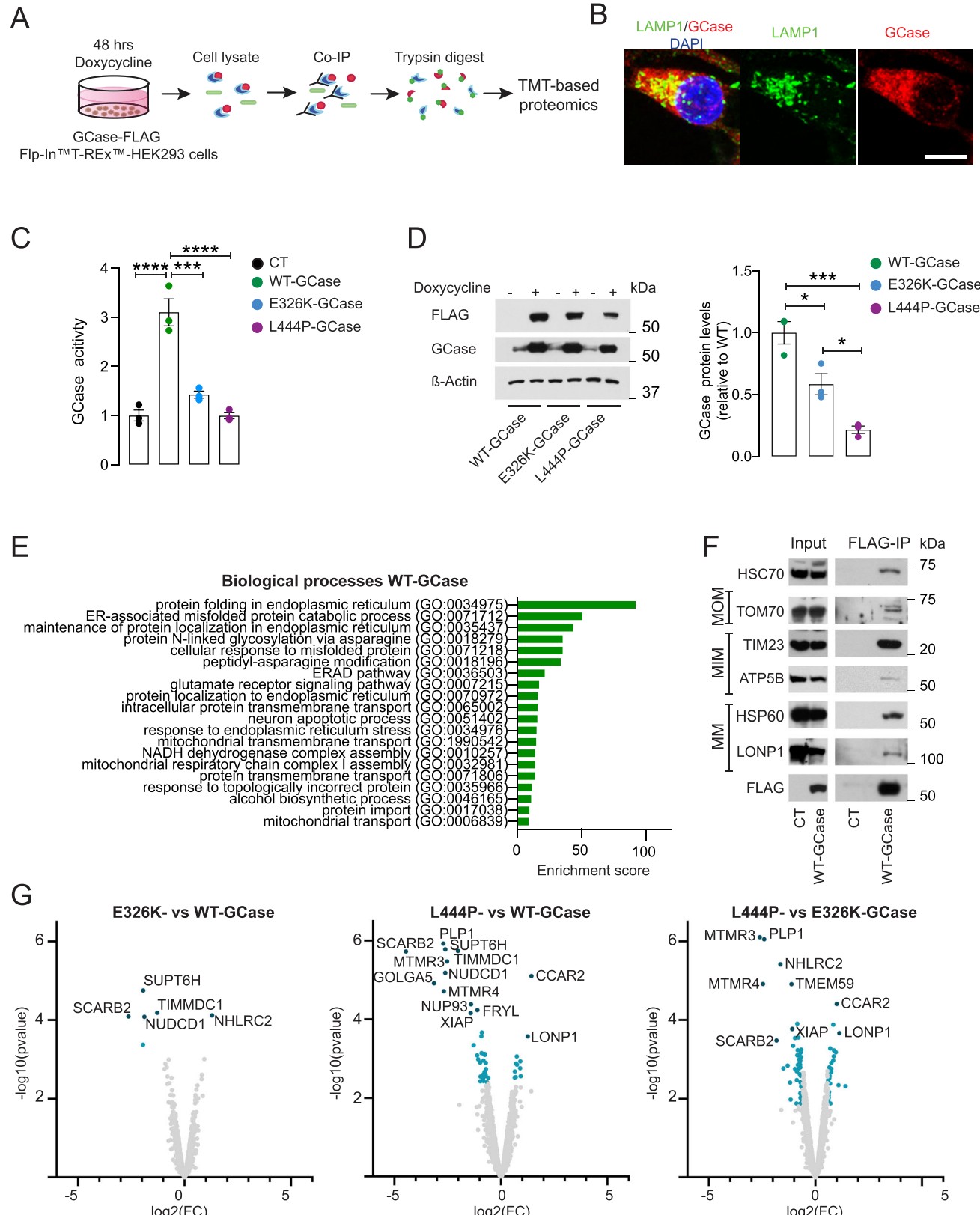

and the correct localization of MTS-GFP$_{1-10}$ was confirmed by assessing the colocalization between LONP1 and MTS-GFP$_{1-10}$ 48 h after doxycycline induction (Fig. 2C). To test whether GCase traffics to mitochondria, the nine aa linker and S11β-GFP were added at the C-terminus to the WT sequence of GCase (GBA1-GFP$_{11}$) (Fig. 2D). As a positive control, we introduced the MTS of Cox8A at the same

location as the V5-Flag-tag of the N-terminus and the S11β-strand of GFP to the C-terminus in WT-GCase (MTS-GCase). Transient over-expression of WT, E326K, or L444P GBA1-β11 in cells expressing MTS-GFP$_{1-10}$ led to GFP colocalization with MitoTracker Red (Fig. 2E). Next, we deleted the iMTS-ls with the highest score predicted by TargetP (dMTS-GCase). While the deletion of iMTS-ls did not affect

**Fig. 1 | TMT proteomics in stable T-Rex HEK cells reveals GCase mitochondrial interactors.** **A** Experimental plan of the TMT-based quantitative proteomic profiling of GCase interactors in T-Rex HEK cells overexpressing V5-Flag-GCase (WT, L444P, or E326K). T-Rex HEK cells expressing the empty vector were used as a control (CT). **B** Representative fluorescence microscopy image showing the intracellular localization of exogenous WT-GCase (V5, red) and lysosomes (LAMP1, green). Scale bar, 10 μm. **C** GCase activity in T-Rex HEK cells expressing WT or mutant GCase (E326K or L444P). Values are normalized to the empty vector control (CT). Mean ± SEM; one-way ANOVA with Bonferroni post hoc test; ****$P < 0.0001$, ***$P = 0.0004$; $n = 3$ independent experiments. **D** Representative western blot for FLAG and GCase in uninduced and induced T-Rex HEK cells overexpressing V5-Flag-GCase (WT, E326K, or L444P). Densitometric quantification of GCase-FLAG in T-Rex

HEK cells expressing WT or mutant GCase (E326K or L444P) is shown on the right. Mean ± SEM; one-way ANOVA with Bonferroni post hoc test; ***$P = 0.0009$, *$P = 0.0218, 0.0371$, in sequence; $n = 3$ independent experiments. **E** Gene ontology pathway analysis showing the top enriched biological processes (BPs) in the WT-GCase interactome. **F** Representative GCase-FLAG CoIP showing the interaction between GCase and HSC70, TOM70, TIM23, ATP5B, HSP60, and LONP1. MOM mitochondrial outer membrane, MIM mitochondrial inner membrane, MM mitochondrial matrix. **G** Volcano plot showing differentially expressed protein interactors among WT-, E326K-, and L444P-GCase lines. Hit interactors that differ by twofold and have a false discovery rate (FDR) of ≤5% are shown using Limma-based differential analysis. Source data are provided as a Source Data file.

GCase stability (Fig. 2F), it abolished GCase mitochondrial localization, supporting its role in the mitochondrial import of GCase (Fig. 2E).

## GCase is imported into mitochondria by the mitochondrial import machinery

Most mitochondrial proteins are imported by the mitochondrial protein import machinery (TOM, TIM23, mitochondrial HSP70). Upon translocation through the TOM channel, proteins are transferred to the presequence translocase of the inner membrane (TIM23 complex)[17]. Based on the interactome analysis, GCase interacts with several components of the mitochondrial import machinery as well as mitochondrial chaperones and proteases (Fig. 3A and Supplementary Dataset 1). Using a CoIP assay in T-Rex HEK cells, we confirmed that both WT and mutant GCase interact with HSC70, a cytosolic chaperone protein belonging to the HSP70 family involved in mitochondrial import[18] (Figs. 1F, 3B). Furthermore, CoIP experiments confirmed the interaction of WT and mutant GCase with TOM70 and TIM23 (Figs. 1F, 3C, D). Interestingly, L444P-GCase showed a significantly higher interaction with TOM70 (Fig. 3C). The GCase interactome revealed an increased interaction of L444P-GCase with the mitochondrial protease LONP1 compared to WT- and E326K-GCase (Fig. 1G and Supplementary Dataset 1). CoIP experiments performed in T-Rex HEK cells confirmed the increased interaction of L444P-GCase with LONP1 compared to WT- and E326K-GCase (Fig. 3E). We also detected a significantly increased interaction between L444P-GCase and the mitochondrial chaperone HSP60 (Fig. 3F and Supplementary Fig. 3C). Furthermore, immunostaining for the mitochondrial protease LONP1 followed by ExM confirmed the interaction between GCase and LONP1 and revealed increased colocalization between GCase and LONP1 in L444P-GCase cells compared to WT-GCase and E326K-GCase cells (Supplementary Fig. 3D). As a control of the overexpression model system, T-Rex HEK cells overexpressing V5-FLAG hexosaminidase B (HexB) were generated. We did not detect any HexB mitochondrial localization, suggesting the absence of a mislocalization of GCase due to an overexpression-related artifact (Supplementary Fig. 3E, F).To investigate the role of HSC70 and TIM23 in the folding and mitochondrial import of GCase, we employed shRNA-mediated knockdown (KD) in T-Rex HEK cells expressing the L444P mutation, which we found to have a more significant impact on the GCase interaction with mitochondrial proteins. Both HSC70 and TIM23 KD led to a significantly decreased interaction of L444P-GCase with LONP1 (Supplementary Fig. 3G–J), supporting their role in the mitochondrial import of GCase.

## GCase localizes to mitochondria and interacts with mitochondrial chaperones and proteases in human iPSC-derived neuronal cells

To confirm these results at the endogenous level in a disease-relevant model, we generated and characterized a set of isogenic human induced pluripotent stem cells (iPSCs) carrying homozygous and heterozygous L444P GBA1 mutations, E326K heterozygous GBA1 mutations, an artificial severe recombinant variant containing the

p.L444P mutation (RecNcil), and corresponding isogenic controls (Supplementary Fig. 4A–C and Supplementary Table 1). To confirm the mitochondrial localization of endogenous GCase, lysates from isogenic iPSC-derived neural precursor cells (NPCs) generated from L444P/L444P GBA1 iPSCs and isogenic controls were subjected to subcellular fractionation followed by digitonin permeabilization and Triton X-100 solubilization of mitochondrial inner and outer membranes. Western blot analysis confirmed the intramitochondrial localization of endogenous GCase (Supplementary Fig. 4D). Next, we sought to validate the interaction of GCase with mitochondrial proteins identified in the interactome analysis. To this end, lysates from isogenic iPSC-derived neural precursor cells generated from L444P/L444P GBA1 iPSCs and isogenic controls were subjected to CoIP using an antibody against GCase, which was first validated using GBA1 knockout iPSC-derived NPCs and a known interactor, LIMP2 (Supplementary Fig. 4E, F). Western blot analysis confirmed the interaction of GCase with the mitochondrial proteins LONP1, HSP60, TIM23, NDUFS2, and TIMMDC1 (Fig. 3G). Furthermore, to validate the mitochondrial localization of GCase at the endogenous level, isogenic NPCs were differentiated into midbrain dopaminergic (DA) neurons (Supplementary Fig. 5A). No significant differences in the iPSC differentiation potential were observed between mutants and controls (Supplementary Fig. 5B). Immunofluorescence images showed an increase in the mitochondrial localization of mutant GCase (Supplementary Fig. 5C–F). Moreover, higher-resolution ExM immunofluorescence images confirmed the increased colocalization between L444P-GCase and LONP1 in iPSC-derived DA neurons (Fig. 3H).

## GCase maintains OXPHOS complex I integrity in HEK cells and iPSC-derived dopaminergic neurons

Whole-cell lysate TMT proteomic analysis revealed dysregulation of mitochondrial CI pathways (Supplementary Fig. 2C). Furthermore, the GCase interactome showed a decrease in the interaction between E326K- and L444P-GCase and TIMMDC1, which is involved in mitochondrial CI assembly[19] (Fig. 1G). CoIP followed by western blot analysis confirmed the decreased interaction between E326K- and L444P-GCase and TIMMDC1 compared to WT-GCase in T-REX cells (Fig. 4A, B). Similarly, the interaction between GCase and NDUFA10, a CI subunit, was significantly reduced in E326K- and L444P-GCase cells compared to WT-GCase cells (Fig. 4A, B). Interestingly, overexpression of WT- and E326K-GCase led to an increase in the CI subunits NDUFS1 and NDUFA9 in T-REX cells (Supplementary Fig. 6A). No significant difference was observed in L444P-GCase compared to CT cells. In contrast, NDUFS1 and NDUFA9 levels were significantly lower in L444P- than WT- and E326K-GCase cells (Supplementary Fig. 6A). To determine whether GCase plays a role in maintaining the higher-order integrity of CI and its supercomplexes (also named respirasomes or SC I + III2 + IV), we employed Blue native-PAGE followed either by CI in-gel activity assay (IGA) or by immunoblotting to detect its subunit NDUFA9 in WT-, E326K-, and L444P-GCase T-REX cells. The practical totality of fully assembled active CI was localized in association with the respirasomes, which were significantly increased in WT- and E326K-GCase cells

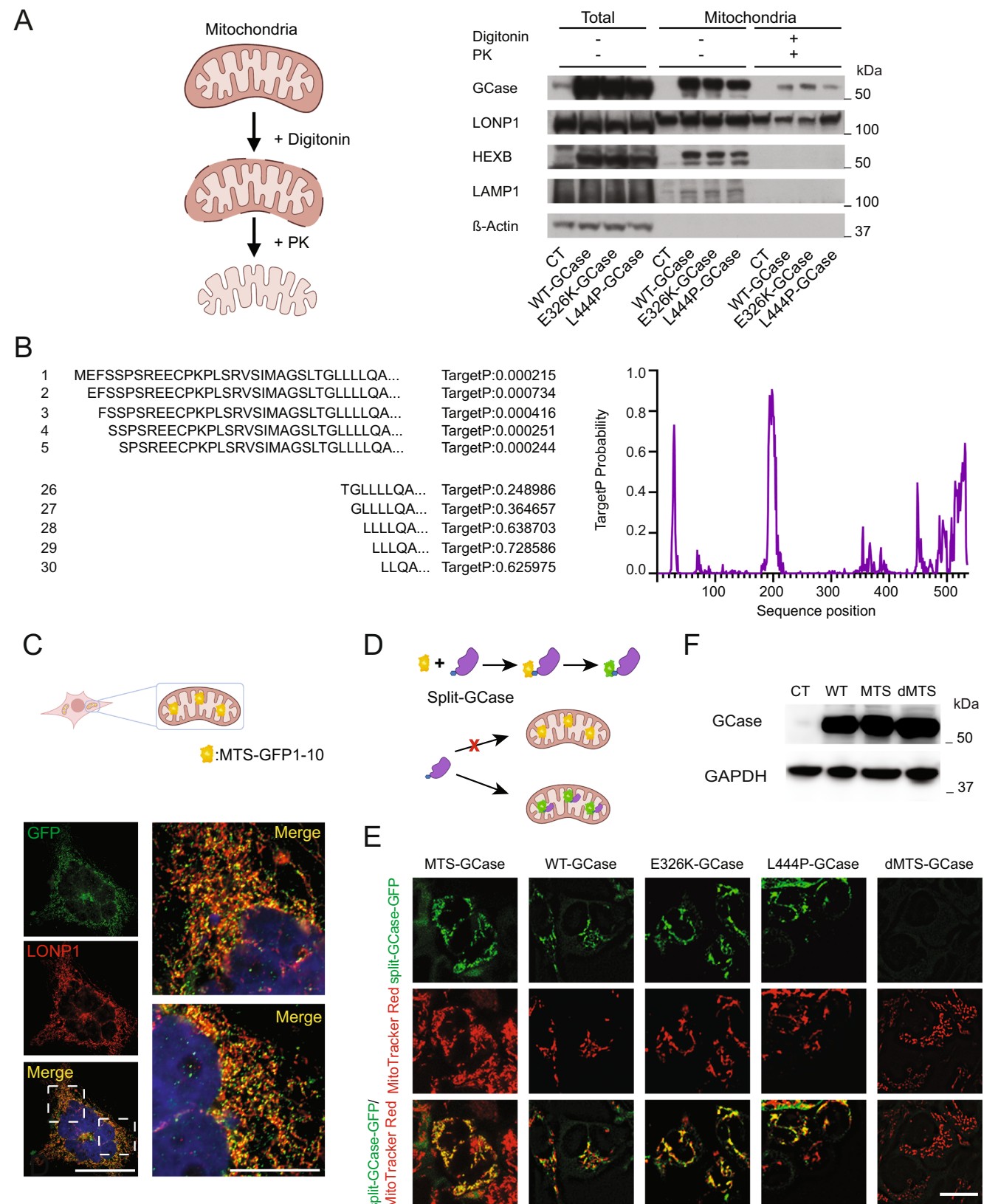

compared to the control conditions (Fig. 4C, D). However, a significant decrease in respirasome levels was observed in L444P-GCase cells compared to WT cells (Fig. 4C, D). BN gels also showed a decrease in TIMMDC1 complexes at ~400/440 kDa, which are involved in ND1-module assembly[19], in mutant GCase cells (Supplementary Fig. 6B). The higher-molecular-weight species of >800 kDa, which contain both

MCIA and TIMMDC1, were also significantly reduced (Supplementary Fig. 6B). Next, we performed BN gels to visualize intact CI and its supercomplexes in NPCs and DA neurons differentiated from L444P/L444P and E326K/WT iPSCs and their isogenic controls as well as *GBA1* KO iPSCs. In this model, the totality of fully assembled CI again localized within the respirasomes in both NPCs and DA neurons

**Fig. 2 | GCase is targeted to mitochondria. A** Isolated mitochondria from the control, WT-, E326K-, and L444P-GCase lines were solubilized with digitonin and subjected to digestion with proteinase K (PK), and immunoblotting was performed for the indicated markers. Images were created with BioRender.com. **B** TargetP probability scores of the internal mitochondrial target sequence-like sequence (iMTS-ls) of the sequences resulting from amino acid removal. To calculate the scores, we consecutively N-terminally truncated the sequences and calculated the corresponding TargetP scores for each position. **C** Schematic representation of mitochondrial targeting of the MTS-GFP$_{1-10}$ split-GFP system and representative immunofluorescence image showing the colocalization of LONP1 with MTS-GFP$_{1-10}$

48 h after doxycycline induction. Scale bar, 10 μm. Images were created with BioRender.com. **D** Schematic representation of the MTS-GFP$_{1-10}$ split GCase GFP system. Images were created with BioRender.com. **E** Confocal images from MTS-GFP$_{1-10}$ T-Rex HEK cells transfected with WT, E326K, or L444P GBA1-GFP$_{11}$ or dMTS-GBA1. Representative fluorescence microscopy images showing the intracellular distribution of GFP (green) and mitochondria stained with MitoTracker Red. Scale bar, 10 μm. **F** Representative western blot showing GCase levels in T-Rex HEK cells overexpressing the empty vector (CT), WT-GCase, MTS-GCase, or dMTS-GCase. Source data are provided as a Source Data file.

(Fig. 4E). The amount of respirasomes was decreased in *GBA1* KO and GBA1 mutant NPCs and DA neurons, suggesting a role for GCase in the structural maintenance of CI (Fig. 4E, F).

## GCase modulates CI activity and respiration in HEK cells and iPSC-derived neurons

Next, we sought to further investigate the impact of GCase on CI activity. To this end, we first employed WT-, E326K-, and L444P-GCase T-Rex HEK cells. Interestingly, overexpression of GCase led to an increase in CI activity, with a significant reduction in E326K- and L444P-GCase compared to WT-GCase cells (Supplementary Fig. 7A). The deletion of the iMTS-ls led to a decrease in CI activity compared to WT-GCase lines, without affecting Complex IV (CIV) activity (Supplementary Fig. 7B). In parallel, we measured the oxygen consumption rate (OCR) in WT-, L444P-, and E326K-GCase T-Rex HEK cells. Similar to that observed for CI activity, overexpression of GCase led to an increase in basal, ATP-linked, and maximal respiration (Supplementary Fig. 7C). However, the OCR was significantly decreased in L444P and E326K cells compared to WT cells (Supplementary Fig. 7C). We validated these results in DA neurons generated from *GBA1* iPSCs and their corresponding isogenic controls. CI activity was significantly decreased in L444P and E326K iPSC-derived neurons compared to the corresponding isogenic controls (Fig. 4G). The activity of CII and CIV was also reduced in mutant neurons but the difference was not significant (Supplementary Fig. 7D). Moreover, the OCR was significantly decreased in L444P and E326K neurons compared to WT cells (Supplementary Fig. 7E).

## GBA1 mutant midbrain organoids show alterations in CI function and perturbations in energy metabolism pathways

To investigate GBA1-related pathways in a model that better resembles the complexity of the human disease, *GBA1* mutant iPSC lines and corresponding isogenic controls were differentiated into midbrain organoids. To improve midbrain organoid differentiation, we combined a previous method that promotes midbrain floor-plate formation[20] with a recent strategy based on the biphasic activation of WNT signaling, which promotes the reproducible 2D differentiation of midbrain DA neurons from pluripotent stem cells[21] (Fig. 5A). At 18 days in vitro (DIV), embryoid bodies showed uniform neuroectoderm formation (Fig. 5B). Developing organoids showed tubular structures containing midbrain progenitors expressing OTX2 (Fig. 5C and Supplementary Fig. 8A). Midbrain organoids derived from *GBA1* mutant iPSCs and corresponding isogenic controls displayed similar sizes and organization (Figs. 5B, C and Supplementary Fig. 8B, C). Cells expressing tyrosine hydroxylase (TH), the rate-limiting enzyme in DA synthesis, were observed at 35 DIV, and the number and complexity of cellular processes increased over time (Fig. 5D). Long-term culture midbrain organoids showed DA neuron maturation and the formation of neuromelanin starting at 70 DIV, with a peak at 120 DIV (Supplementary Fig. 8D–F). We observed S100B-positive cellular processes starting 35 days after induction (Supplementary Fig. 8G). Furthermore, *GBA1* organoids showed a significant reduction in GCase activity and protein levels compared to isogenic controls at 42 DIV (Supplementary Fig. 9A, B). *GBA1* mutant organoids showed a

significant time-dependent reduction in TH + neurons compared to isogenic controls (Fig. 5E, F). The levels of insoluble A-SYN were increased in mutant organoids compared to isogenic controls (Fig. 5G and Supplementary Fig. 9C). Furthermore, p-Ser129-A-SYN neurons were observed at 100 DIV in GBA1 mutant organoids (Fig. 5H).

To gain insight into cell-type-specific mechanisms, we transcriptionally profiled *GBA1* mutant midbrain organoids and gene-corrected controls at the single-cell level using 10X Genomics Chromium scRNA-seq. Based on the evidence of significant changes in A-SYN levels and comparable levels of DA neurons (Fig. 5E–G), organoids were analyzed at 42 DIV. To validate midbrain organoid differentiation and reproducibility, we initially profiled individual organoids from different batches (Supplementary Fig. 10A–C). In subsequent experiments, we pooled five organoids/line per experiment. Upon quality control (Supplementary Fig. 10D), normalization and integration were performed to reduce the batch effect and compare the genotypes (Supplementary Fig. 10E). We identified 21 cell clusters present in all lines in both mutant and gene-corrected iPSC-derived organoids (Supplementary Dataset 3). After cluster annotation, subclusters were merged into five main clusters (Fig. 6A), namely, progenitor cells such as radial glia-like cells (*MSX1, PAX3, SOX2,* and *RFX4*) as well as neural progenitor cells (*NEUROG1* and *NEUROD4*), neuronal (*DCX* and *NEFM*), astrocyte (*TNC* and *AQP1*), and oligodendrocyte (*SOX10* and *APOD*) clusters (Fig. 6A–C). Residual floor-plate progenitor markers were found within the radial glia cluster (Supplementary Fig. 10F). Overall, the cell-type composition was comparable among mutant and isogenic control organoids (Fig. 6B). To dissect the chronological development of midbrain organoids, we analyzed organoids at 42, 65, and 100 DIV and constructed cell fate trajectories by using the lineage decision method Monocle 3 (Supplementary Fig. 11A, B). The trajectory proceeded along two branches toward cells expressing radial glia markers (*SOX2, RFX4*) or neural progenitor markers (*NEUROG1* and *NEUROD4*) (Supplementary Fig. 11C). At 100 DIV, the trajectories led to two distinct states, oligodendrocytes and astrocytes (*SOX10, APOD, AQP1,* and *S100B*) as well as pericytes (*PDGFRB* and *DCN*) (State 1) and neurons expressing high levels of *DCX, EN1,* and *LMO3* (State 2) (Supplementary Fig. 11C, D). Interestingly, such a late emergence of pericyte clusters has also been described in the developing human midbrain[22]. Confirming neuronal maturation, cells belonging to State 2 expressed high levels of *CALB1, SYP, SNCA, GAD1,* and *TH* (Supplementary Fig. 11C, D). To assess the similarity between midbrain organoid generation protocols, we employed a recently published scRNA-seq dataset[23], and we found a high similarity between floor-plate markers along the radial glia clusters as well as a high similarity of neuronal markers and DA markers (Supplementary Fig. 11E, F). Interestingly, while the reference dataset showed an abundance of radial glial cells, our dataset displayed enrichment in neuronal clusters at 42 DIV (Supplementary Fig. 11G). To dissect the impact of *GBA1* mutations on specific neuronal populations in our dataset, we subclustered neuronal populations in eight neuronal clusters, including two distinct DA neuronal clusters, two GABAergic (*GAD1, GAD2,* and *SLC32A1*) and two glutamatergic (*SLC17A6* and *GRIN2B*) neuronal clusters and an undefined neuronal cluster (*DCX,*

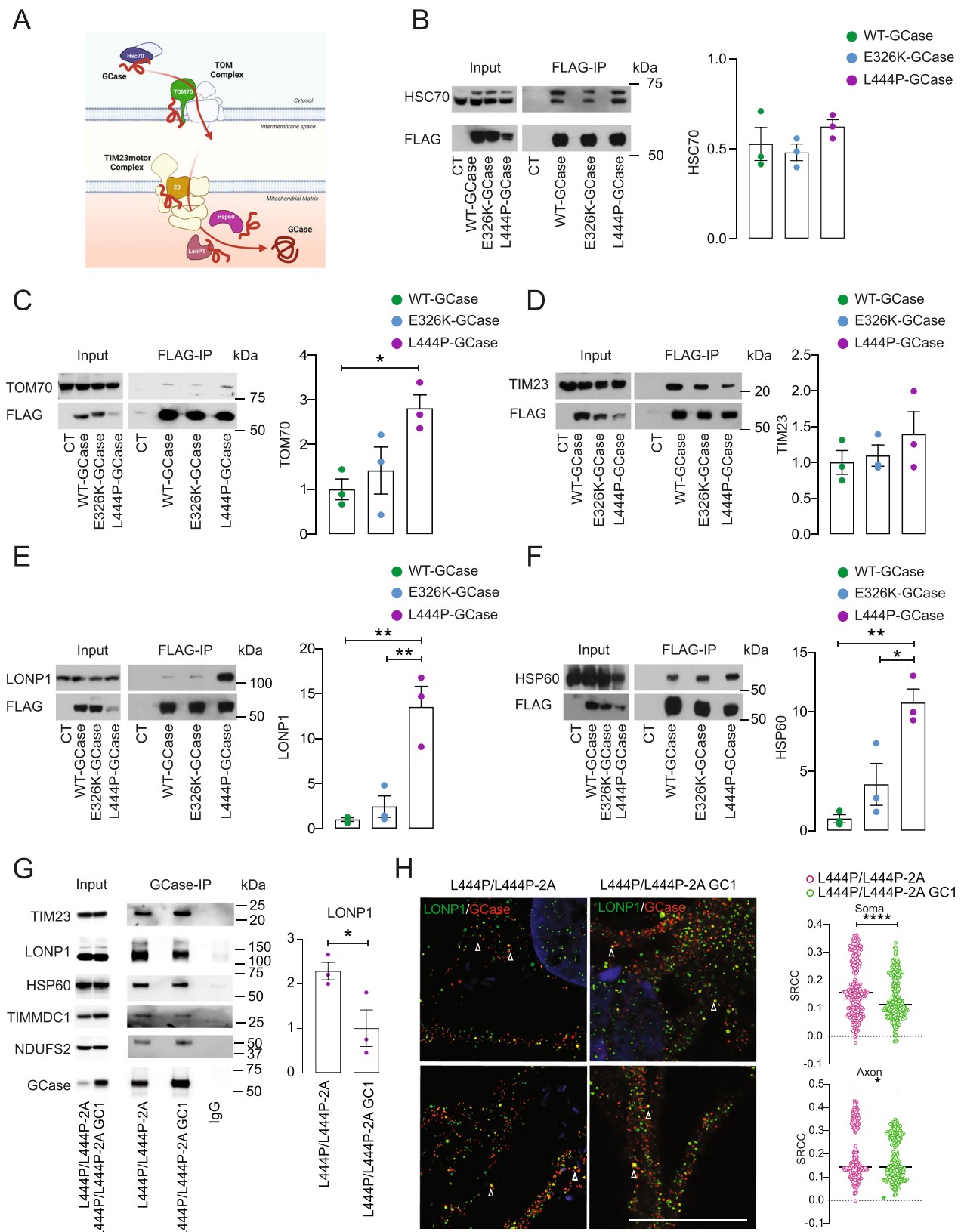

*SYP*, and *VAMP2*) (Supplementary Fig. 12A, B). Synaptic transcription factor genes were highly expressed within the neuronal clusters (Fig. 6D and Supplementary Fig. 12C). Differential gene expression and pathway analysis of neuronal clusters in scRNA-seq data revealed enrichment of metabolic pathways across isogenic pairs, with a stronger impact on mitochondrial ATP-related pathways in the severe

mutant recombinant L444P *GBA1* organoids (Fig. 6E and Supplementary Fig. 12D). In this respect, pathway analysis revealed a selective dysregulation of mitochondrial energy metabolism in DA neuron clusters from *GBA1* mutant organoids compared to isogenic controls (Fig. 6E and Supplementary Fig. 12D), suggesting that severe *GBA1* mutations drive selective mitochondrial disturbances in DA neurons.

**Fig. 3 | GCase is imported into mitochondria by the mitochondrial protein import machinery and interacts with mitochondrial chaperones and proteases. A** Graphical summary showing cytosolic chaperones as well as mitochondrial import and matrix GCase interactors identified by TMT proteomics. Images were created with BioRender.com. **B–D** Validation of WT and mutant (E326K and L444P) GCase interactions with HSC70 (**B**), TOM70 (**C**), and TIM23 (**D**) by coimmunoprecipitation followed by western blot analysis and densitometry quantification. Mean ± SEM; one-way ANOVA with Bonferroni post hoc test, *$P = 0.0432$; $n = 3$ independent experiments. **E, F** Validation of WT and mutant (E326K and L444P) GCase interactions with LONP1 (**E**) and HSP60 (**F**) by coimmunoprecipitation followed by western blot analysis and densitometry quantification. Mean ± SEM; one-way ANOVA with Bonferroni post hoc test, **$P = 0.0031$, 0.0059, 0.0041, in sequence; *$P = 0.0225$; $n = 3$ independent experiments. **G** Immunoprecipitation of endogenous GCase followed by western blot analysis demonstrating GCase interaction with mitochondrial proteins LONP1, HSP60, TIM23, TIMMDC1, and NDUFS2 in NPCs generated from L444P/L444P iPSCs and the corresponding gene-corrected isogenic control. Quantification of the LONP1/GCase interaction in L444P/L444P iPSC-derived NPCs and isogenic controls is shown on the right. Mean ± SEM; unpaired two-tailed $t$-test, *$P = 0,048$, $n = 3$ independent experiments. **H** ExM images showing LONP1 (green) and GCase (red) staining in L444P/L444P neurons and corresponding isogenic gene-corrected controls. Scale bar, 100 µm. Colocalization analysis of LONP1/GCase was performed in the neuronal soma and axons (**H**). Mean ± SEM; unpaired two-tailed $t$-test; ****$P < 0.0001$, *$P = 0.0275$. Five images were taken from $n = 3$ independent experiments. Source data are provided as a Source Data file.

Such selective impairment of metabolism in DA neurons was not evident in E326K and L444P mutant organoids. However, more than 50% of the neuronal mitochondrial DEGs in all isogenic couples belonged to respiratory chain complexes (Supplementary Dataset 4). In line with our findings in stable HEK cell lines and iPSC-derived neurons, CI activity was significantly reduced in *GBA1* mutant organoids compared to isogenic controls (Fig. 6F). To further validate the findings of the interactome analysis in midbrain organoids; we performed immunostaining for the mitochondrial protease LONP1 followed by ExM in the L444P-GCase mutant and gene-corrected controls, which confirmed the interaction between GCase and LONP1 and revealed increased colocalization between GCase and LONP1 in L444P/L444P organoids (Fig. 6G).

**Mitochondrial LONP1 protease contributes to the degradation of mutant GCase**

Proteome, CoIP, and ExM experiments showed that the mitochondrial LONP1 protease interacts with GCase and that *GBA1* mutations significantly increase this interaction. Therefore, we sought to dissect the role of LONP1 in the regulation of mitochondrial GCase. To this end, we treated T-Rex HEK cells overexpressing V5-Flag-GCase (WT, L444P, or E326K) with noncytotoxic concentrations of 2-cyano-3,12-dioxooleana-1,9-dien-28-oicacid (CDDO), which inhibits LONP1 protease[24] but not the 26 S proteasome[25]. As expected, CDDO treatment induced a significant upregulation of the mitochondrial unfolded protein response (UPR^mt) (Supplementary Fig. 13A). No significant changes in the expression level of UPR^mt genes were observed when comparing WT, L444P, and E326K cells (Supplementary Fig. 13A). Interestingly, CDDO treatment caused a significant reduction in CI activity in WT-GCase T-Rex cells, whereas mutant GCase cells displayed a significant increase in CI activity upon CDDO treatment (Supplementary Figure 13B). Given the link between GCase activity and A-SYN[26] and the role of LONP1 in regulating A-SYN pathology[27], we treated mutant and isogenic control DA neurons with Alexa Fluor 594-labeled A-SYN preformed fibrils (PFFs) and assessed A-SYN localization by live-cell imaging in the presence or absence of CDDO. E326K and L444P *GBA1* mutant neurons showed increased levels of intracellular A-SYN PFFs compared to isogenic controls, suggesting increased uptake or defects in the degradation of A-SYN (Fig. 7A and Supplementary Fig. 13C). Upon CDDO treatment, we observed a significant increase in intracellular A-SYN PFFs in isogenic control neurons, whereas E326K and L444P *GBA1* did not show significant changes (Fig. 7A). To assess selective PFF accumulation at mitochondria, we treated iPSC-derived neurons with CDDO prior to incubation with A-SYN PFFs, and we performed live-cell imaging with MitoTracker. CDDO treatment led to an increase in A-SYN PFFs colocalizing with mitochondria in isogenic control and E326K mutant iPSC-derived neurons (Fig. 7A). Furthermore, we observed a significant decrease in CI activity in isogenic control neurons upon CDDO treatment; whereas a nonsignificant increase was observed in mutant neurons (Fig. 7B).

## Discussion

Here, we identified a function of GCase beyond the lysosome, which links this disease-relevant lysosomal protein to mitochondria. Our data provide clear evidence that GCase is associated with mitochondria. We show that the HSC70/TOM70 pathway mediates the mitochondrial import of GCase. TOM70 is an essential component of the import machinery of mitochondrial proteins with iMTS-ls[28], and cytosolic ATP-dependent HSC70/HSP70 chaperones play a key role in preprotein targeting to mitochondria[18]. Interestingly, we identified an increased interaction between mutant GCase and proteins of the mitochondrial protein quality control system, namely, HSP60 and LONP1. Most pathogenic *GBA1* mutations are missense mutations, and the mutant enzyme has sufficient residual enzymatic activity when properly folded and trafficked to the lysosome. Hence, it is possible that the increased interaction of mutant GCase with HSP60 and LONP1 reflects a mitochondria-based quality control system for cytoplasmic proteins[29,30]. Indeed, it has been shown that aggregation-prone proteins are partially imported and degraded by mitochondria[29]. Furthermore, cytosolic misfolded proteins are associated with and can be transported into mitochondria[31]. Based on this evidence, mutant GCase would be transported and degraded within mitochondria[29]. It is also interesting to note that *GBA1* mutations did not cause significant upregulation of UPR^mt genes. In addition to its proteolytic activity, LONP1 also shows an intrinsic chaperone-like function and cooperates with mitochondrial HSP70 in the folding pathway in vitro[32]. Our results suggest that under normal conditions, LONP1 might function, to a larger degree, in GCase folding within mitochondria. The severe L444P variant has a significantly higher impact on the degree of GCase/LONP1 interaction compared to the E326K variant, suggesting a stronger activation of the chaperone and proteolytic activity of this mitochondrial protease. In the presence of *GBA1* mutations, the increased chaperone and proteolytic LONP1 activity may therefore promote the degradation of misfolded/damaged intramitochondrial GCase. LONP1 inhibition ameliorates CI activity in *GBA1* mutant HEK cells, while the opposite effect is observed in WT-GCase cell models. Similar results were obtained in iPSC neurons, even though the increase in CI activity upon LONP1 inhibition in *GBA1* mutant cells was not significant. CDDO-Me has been shown to specifically inhibit the ATP-hydrolase proteolytic function of LONP1[25]. Furthermore, studies in proteolytically inactive yeast LONP1 homologous show that an increase in chaperone function could promote mitochondrial membrane complex assembly[33]. These data support the hypothesis that in the presence of mutant GCase, LONP1 proteolytic activity is more pronounced than its chaperone function. The increased interaction between mutant GCase and LONP1 may also interfere with LONP1 folding properties leading to mitochondrial protein aggregation, which can also partially explain the decrease in CI activity and A-SYN mitochondrial accumulation in iPSC-derived neurons (Fig. 7C). Given the low resolution of confocal microscopy; we are unable to define the exact localization of exogenous A-SYN PFFs. Interestingly, recent

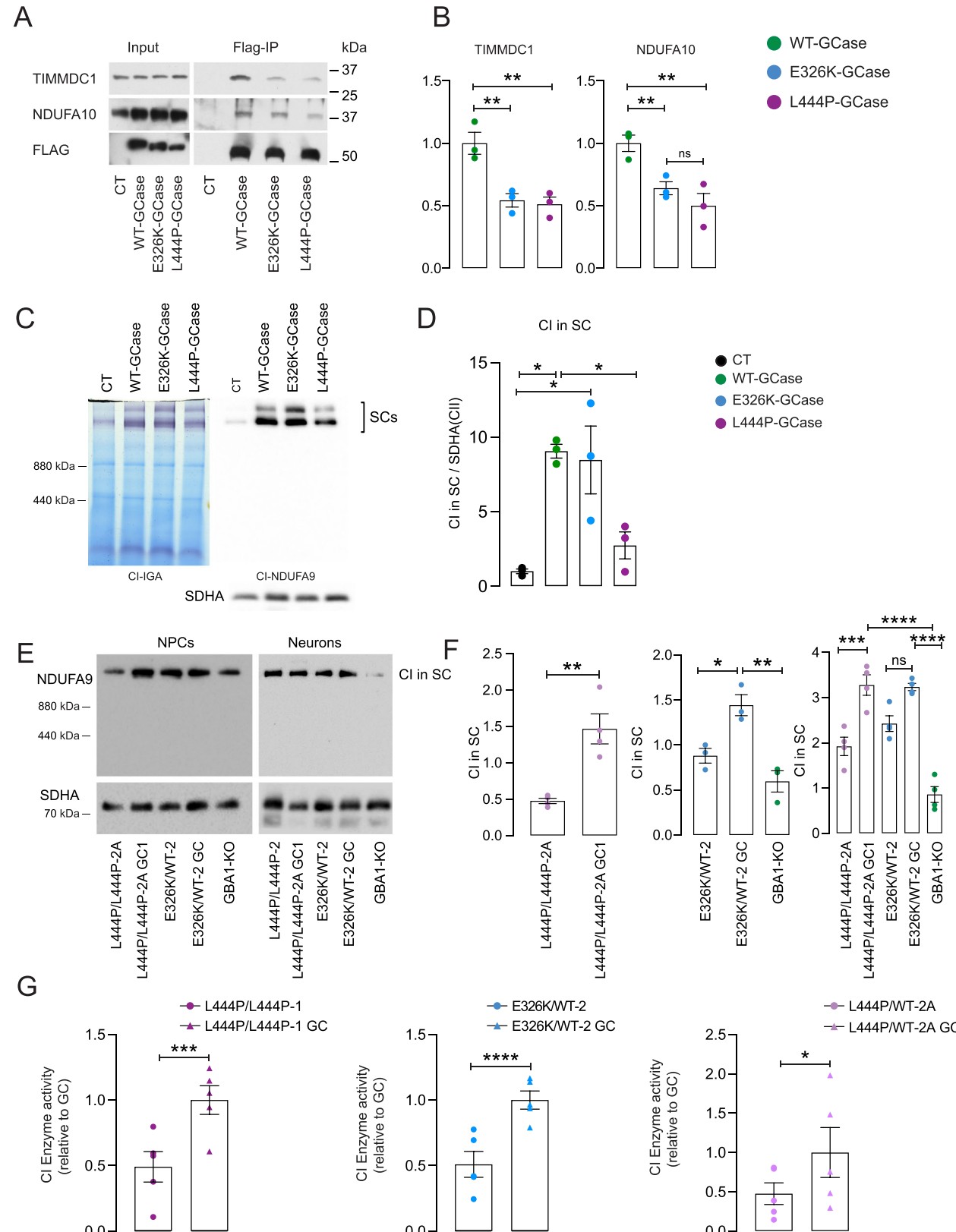

work has shown that intracellular seeding events occur preferentially on membrane surfaces, especially at mitochondrial membranes[34]. As mitochondrial proteostasis can influence the homeostasis of cytosolic aggregation-prone proteins[27], one hypothesis could be that LONP1 regulates A-SYN pathology at mitochondrial membranes. It is also interesting to note that LONP1 inhibition did not cause an increase in A-SYN PFF levels at mitochondria in iPSC-derived neurons carrying the severe L444P mutation. These results suggest that the burden of misfolded GCase may overload the capacity of mitochondrial proteases. The regulatory mechanisms controlling the interaction between LONP1 and folding GCase substrates versus GCase proteolytic substrates should be further explored.

**Fig. 4 | GCase modulates mitochondrial CI integrity and activity in iPSC-derived neurons. A** FLAG-GCase-CoIP and western blot analysis showing the interaction between GCase and TIMMDC1 and NDUFA10 in T-Rex HEK cells overexpressing V5-Flag-GCase (WT, L444P, or E326K). **B** Densitometric quantification of the interaction between GCase and TIMMDC1 or NDUFA10. Mean ± SEM; one-way ANOVA with Bonferroni post hoc test; $n = 3$ independent experiments; TIMMDC1: **$P = 0.0098$, 0.0070, in sequence; NDUFA10: **$P = 0.0097$, *$P = 0.0439$. **C** Blue native and western blot analysis of complex I (CI) assembly into the mitochondrial supercomplex (SC) in T-Rex HEK cells overexpressing V5-Flag-GCase (WT, L444P, or E326K). Succinate dehydrogenase complex flavoprotein A (SDHA) was used as a loading control. **D** Densitometric quantifications of the CI-containing supercomplex in the different cell lines are shown on the right. Data were normalized by CII (SDHA). Mean ± SEM; one-way ANOVA with Bonferroni post hoc test, *$P = 0.011$, 0.0172, 0.0429, in sequence; $n = 3$ independent experiments. **E** Blue native analysis of CI

assembly into the mitochondrial supercomplex (SC) in NPCs and DA neurons differentiated from L444P/L444P and E326K/WT iPSCs and their isogenic controls as well as *GBA1* KO iPSCs. Succinate dehydrogenase complex flavoprotein A (SDHA) was used as a loading control. **F** Densitometric quantifications of the CI-containing supercomplex in the different cell lines are shown on the right. Data were normalized by CII (SDHA). Mean ± SEM; left panel: **$P = 0.0032$, unpaired two-tailed *t*-test; middle panel: *$P = 0.0291$, **$P = 0.004$, one-way ANOVA with Bonferroni post hoc test; right panel: ***$p = 0.0008$, ****$p < 0.0001$; one-way ANOVA with Bonferroni post hoc test; $n = 4$ independent experiments. **G** CI activity in enriched mitochondria from isogenic *GBA1* mutant (E326K, L444P) and GC iPSC-derived neurons. Data were normalized to protein content and expressed relative to the isogenic control. Mean ± SEM; two-tailed *t*-test; ***$P = 0.0008$, ****$P < 0.0001$, *$P = 0.0486$; $n = 5$ independent experiments. Source data are provided as a Source Data file.

As LONP1 is also involved in the folding of CI subunits as well as the degradation of CI subunits in depolarized mitochondria[32,35], its overload by mutant GCase may further contribute to the observed CI instability and mitochondrial energy defects. Our results suggest that, under physiological conditions, GCase plays a functional role in regulating energy cellular homeostasis by interacting with TIMMDC1 and by promoting mitochondrial CI integrity. Interestingly, findings from Guarani et al. already indicated a possible interaction between GCase and TIMMDC1 in human cell lines[19]. TIMMDC1-associated CI assembly defects have also been linked to a decrease in mature CI[19,36]. We were unable to identify the characteristic accumulation of CI assembly intermediates that commonly occur upon disturbance of CI assembly. The increased binding of mutant GCase to mitochondrial proteases involved in quality control, together with the decreased levels of TIMMDC1 complexes in the mutants, led us instead to deduce that CI integrity is compromised. These results thus suggest that while GCase may not directly contribute to CI assembly, it modulates intermediate stages of the assembly process involving TIMMDC1. GD and PD mutations negatively interfere with the interaction between GCase and CI subunits. This would explain the decreased CI activity, which has been observed in both GCase-deficient and mutant models[37,38]. Both variants affect CI stability and function as well as cellular respiration, with the L444P variant having a stronger, although not significant, impact than the E326K variant. These data suggest a major contribution of CI and mitochondria to *GBA1*-driven neurodegeneration.

Impairment of mitochondrial metabolism and dynamics has been observed in vivo as well as in a variety of cellular models of GCase deficiency and patient neurons[37–40]. Possible mechanisms linking GBA1 to mitochondrial dysfunction include autophagy (mitophagy), ER stress, lipid disturbances, and A-SYN aggregation. Our data suggest that GCase may also play a direct role in the maintenance of CI integrity. Several studies suggest mitochondrial impairment in PD, although it is still unclear whether such mitochondrial defects are primary causes or secondary effects of neurodegeneration. A recent study showed that CI dysfunction may drive PD degenerative processes and clinical phenotypes in a mouse model[41] and this observation is consistent with the finding of significant and specific CI deficiency in the human PD substantia nigra[42,43]. Our data further support the hypothesis that CI dysfunction may be a key pathogenetic driver of PD. From a therapeutic perspective, our data point toward increasing GCase levels and modulating the mitochondrial quality control system as potential therapeutic approaches in *GBA1*-derived neurodegeneration. Due to the decrease in GCase activity that has been observed in sporadic cases, patients affected by dementia with Lewy bodies, and aged individuals[44–46], such a mechanism could be of broad therapeutic relevance. In summary, our studies link GCase toxicity directly to mitochondrial bioenergetics and propose targeting GCase mitochondrial localization as a promising therapeutic approach for neurodegeneration.

## Methods

### Ethics approval for experiments involving human samples
All cells used in the study were derived from patients who signed an informed consent form. The ethics committee of the Medical Faculty at the University Hospital Tübingen (Ethikkommission der Medizinischen Fakultät am Universitätsklinikum Tübingen) approved the protocol prior to performing the experiments.

### Constructs and generation of stable cell lines
Constructs were purchased from IDT (gBlocks Gene Fragments) and subcloned into pcDNA™5/FRT/TO (Thermo Fisher Scientific, # V652020). For the generation of V5-FLAG-GCase lines, the tag was positioned at the N-terminus, three aa after the cleavage site of the leader sequence. These three amino acids are repeated after the tag to ensure that the *GBA1* sequence is intact. To ensure proper cleavage, the V5-FLAG tag was inserted 12 bp after the cleavage site, and this 9 bp were repeated after the tag. To avoid interference with proper protein folding, we employed the short V5 sequence (IPNPLLGLD). Site-directed mutagenesis was performed according to the manufacturer's protocol (Agilent, QuickChange II XL), and base pair exchange was confirmed by Sanger sequencing. Flp-In™ 293 T-Rex cells (Thermo Fisher Scientific, # R78007; RRID:CVCL_U427) were grown in media composed of Dulbecco's modified Eagle's medium (DMEM, Sigma−Aldrich), 10% fetal bovine serum (Gibco), 1% GlutaMax (Gibco) supplemented with 100 µg/ml Zeocin (InvivoGen) and 15 µg/ml blasticidin (InvivoGen). Inducible Flp-In T-Rex 293 cells were generated according to the manufacturer's protocol (Thermo Fisher Scientific). The selection was performed with DMEM supplemented with 15 µg/ml blasticidin and 100 µg/ml hygromycin B Gold (InvivoGen) 48 h after transfection and continued until the expression of the gene of interest was induced by treating the cells with 50 ng/ml doxycycline hyclate (Sigma−Aldrich) for 48 h. A list of primer sequences is provided in Supplementary Table 2.

### FLAG coimmunoprecipitation
Cells were detached using Accutase (Sigma−Aldrich) and washed 2x with ice-cold phosphate-buffered saline (PBS, Sigma−Aldrich). Cells were lysed in 1x TBS (pH 7.4) containing 0.5% NP40 and a protease/phosphatase inhibitor cocktail (Pierce, #A32959). Then the lysates were filtered through a 0.22 µm PES syringe filter (Millipore). About 10 mg lysates were immunoprecipitated with prewashed M2 anti-flag agarose resin (Sigma−Aldrich, #A2220) for 2 h at 4 °C while rotating. For Flag coimmunoprecipitation, Flag was eluted with Flag Peptide (Sigma−Aldrich, #F3290) by moving on a wheel for 20 min at 4 °C followed by a spin for 1 min at 1000 × g. The flow-through was collected and subjected to TMT labeling or Western blot analysis. The antibodies are listed in Supplementary Table 3.

### Endogenous coimmunoprecipitation
Protein G agarose fast-flow beads were washed three times. To this end, the beads were incubated with lysis buffer on the spinning wheel

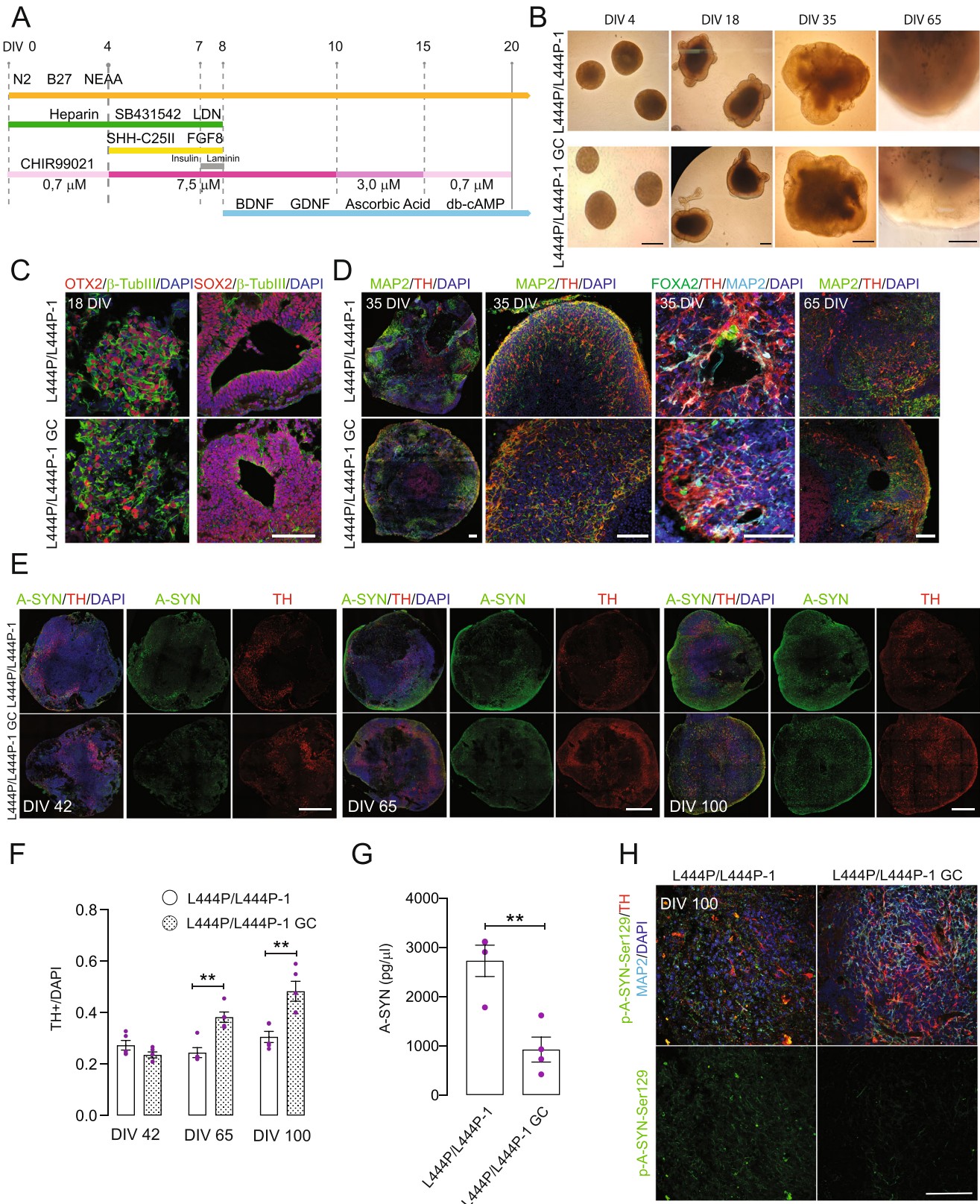

(25 RPM) for 2–5 min at 4 °C, followed by a 1 min centrifugation at 500 RPM 4 °C. After washing the beads, the latter were resuspended in wash buffer and distributed equally to the individual tubes. For the pre-incubation with the antibody against the respective protein, the correct amount of antibody or IgG control were added to the beads: 3 μg anti-Hsp60 antibody and 5 μg of anti-LONP1 antibody were used per IP. As a negative control, the same amount of mouse IgG control (Santa Cruz) and rabbit IgG control (Enzo Life Sciences) were incubated with the beads. The bead antibody mix was incubated for 2 h at 4 °C and 20 RPM on the spinning wheel. After incubating the beads with the antibody, these were washed again 3x. For the CoIP of HSP60 or LonP1, 3.7 ug protein were incubated on the spinning wheel for 2 h with washed antibody decorated beads. Next, the beads were washed three times. Elution was performed by boiling the beads twice with 2x

**Fig. 5 | Generation and characterization of *GBA1* mutant midbrain organoids.**
**A** Schematic representation of the experimental protocol for midbrain organoid generation. **B** Representative brightfield images of midbrain organoids generated from L444P/L444P-1 *GBA1* iPSCs and corresponding isogenic controls. Scale bar, 200 μm. **C** Immunostaining for β-TUBIII (green), SOX2 (red), and OTX2 (red) in 18-day-old midbrain organoids. Cell nuclei were counterstained with DAPI (blue). Scale bar, 100 μm. **D** Immunostaining for MAP2 (green or teal), TH (red), and FOXA2 (green) in midbrain organoids at the indicated timepoints. Cell nuclei were counterstained with DAPI (blue). Scale bar, 100 μm. **E** Immunostaining for A-SYN (green) and TH (red) in midbrain organoids at 42, 65, and 100 DIV. Cell nuclei were counterstained with DAPI (blue). Scale bar, 500 μm. **F** Quantification of TH + neurons in whole-section organoids at 42, 65, and 100 DIV was performed using RapID Cell Counter and expressed as the percentage of DAPI + cells. Mean ± SEM; unpaired two-tailed *t*-test; **P = 0.0011, 0.0038 in sequence; each dot represents the average of cell quantification obtained from three organoids from three different batches. At least three nonsequential cross-section tile images per organoid were quantified. **G** Total A-SYN levels assessed by ELISA in L444P/L444P-1 *GBA1* organoids and corresponding isogenic controls at 42 DIV. Mean ± SEM; unpaired two-tailed *t*-test; **P = 0.0045. Each dot represents an individual midbrain organoid; three independent differentiation onsets. **H** Immunostaining for p-Ser129-A-SYN (green), TH (red), and MAP2 (cyan) in midbrain organoids at 100 DIV. Cell nuclei were counterstained with DAPI (blue). Scale bar, 100 μm. Source data are provided as a Source Data file.

Laemmli buffer. Then the beads were spun for 1 min at 10000 RCF and the supernatant was collected and subjected to Western blot analysis.

## GCase coimmunoprecipitation

Cells were washed 1X with phosphate-buffered saline (PBS, Sigma–Aldrich) and detached using Accutase. The cell suspension was pelleted at 280 rcf for 5′ at 23 °C.

The pellets were then lysed in IP/lysis buffer (Thermo Fisher, #87787) supplemented with a protease/phosphatase inhibitor cocktail (Pierce, #A32959). Coimmunoprecipitation was carried out using the Thermo Fisher Pierce Crosslink Magnetic IP/CoIP Kit (#88805) according to the manufacturer´s instructions. Twenty-five microliters of Pierce protein A/G magnetic beads (Thermo Fisher, #88802-3) were prewashed twice with 1X Modified Coupling Buffer and incubated with 10 μg of GBA MaxPab polyclonal rabbit antibody (Abnova) or normal rabbit IgG (Covalab, #pab01004-P) on a rotating wheel overnight at 4 °C. The following day, the antibody was crosslinked to the beads with a 0.25 mM DSS solution for 1 h on a rotating wheel at RT. Crosslinked magnetic beads were incubated overnight with a total of 7 mg of protein for each lysate. Coimmunoprecipitated proteins were eluted from the beads with 60 μl of the kit-provided Elution buffer pH 2.0 (Pierce, #88805) and neutralized with 6 μl of Neutralization Buffer provided with the kit. Samples were prepared for Western blotting by adding 5x Lane buffer + 10% DTT 1 M to a final concentration of 1X. Each western blot input sample loaded corresponds to a total of 50 μg of protein. Each western blot CoIP sample loaded corresponded to the total of each elution product (66 μl). The antibodies are listed in Supplementary Table 3.

## TMT-based quantitative proteomics

Lysates were pretreated with Benzonase® Nuclease (Sigma) before further processing for LC-MS analysis. Samples were incubated with dithiothreitol at 56 °C for 30 min (10 mM in 50 mM HEPES, pH 8.5) to reduce cysteine. This step was followed by sample alkylation with 2-chloroacetamide (20 mM in 50 mM HEPES, pH 8.5) while protected from light at RT for 30 min. The SP3 protocol was used as published by Hughes et al. for processing the samples[47]. The proteins were digested on beads using trypsin (sequencing grade, Promega) overnight at 37 °C. The ratio between trypsin and proteins was 1:50. TMT11plex Isobaric Label Reagent (Thermo Fisher) was used as described in the manufacturer's protocol. For sample cleanup, an OASIS® HLB μElution Plate (Waters) was used. Six fractions were the outcome of offline high pH reversed-phase fractionation with an Agilent 1200 Infinity high-performance liquid chromatography system equipped with a Gemini C18 column (3 μm, 110 Å, 100 × 1.0 mm, Phenomenex). A trapping cartridge (μ-Precolumn C18 PepMap 100, 5 μm, 300 μm i.d.× mm, 100 Å) and an analytical column (nanoEase™ M/Z HSS T3 column 75 μm × 250 mm C18, 1.8 μm, 100 Å, Waters) were attached to the UltiMate 3000 RSLC nano-LC system (Dionex). For 6 min, a continuous flow of trapping solution (0.05% trifluoroacetic acid in water) onto the trapping column at 30 μL/min allowed peptide trapping. Elution of trapped peptides was achieved by running solvent A (0.1% formic acid in water, 3% DMSO) with a constant flow of 0.3 μL/min through the analytical column, with an increasing percentage of solvent B (0.1% formic acid in acetonitrile, 3% DMSO). The Nanospray Flex™ ion source in positive ion mode allowed direct attachment of the valve of the analytical column to an Orbitrap Fusion™ Lumos™ Tribrid™ Mass Spectrometer (Thermo Fisher Scientific). A Pico-Tip Emitter 360 μm OD × 20 μm ID; 10 μm tip (New Objective) was used to introduce the peptide into the Fusion Lumos using an applied spray voltage of 2.2 kV. The capillary temperature was 275 °C. The resolution of the orbitrap used in profile mode was 120,000, and the mass range was set to 375–1500 m/z during the full mass scan. The maximum filling time was 50 ms. For data-dependent acquisition (DDA), the resolution of the Orbitrap was set to 30,000, and the fill time was set to 94 ms. The ion number was limited to $1 \times 10^5$ ions. A normalized collision energy of 36 was applied. Profile mode was used to acquire MS$^2$ data. The mass spectrometry proteomics data have been deposited to the ProteomeXchange Consortium via the PRIDE partner repository with the dataset identifier PXD032155.

## MS data analysis

IsobarQuant (https://github.com/protcode/isob/archive/1.0.0.zip) and Mascot v2.2.07 (http://www.matrixscience.com/server.html; RRID: SCR_014322) were used for data analysis. The UniProt *Homo sapiens* database (UP000005640, https://www.uniprot.org/proteomes/UP000005640) was used as the reference proteome. The search criteria were determined as follows: carbamidomethyl (C) and TMT11 (K) (fixed modification), acetyl (N-term), oxidation (M), and TMT11 (N-term) (variable modifications). For the full scan (MS1), a mass error tolerance of 10 ppm was set, and a spectrum of 0.02 Da was used for MS/MS (MS2). Two skipped cleavages by trypsin were the maximum permitted. For protein identification, the following criteria had to be fulfilled: (a) at least two distinctive peptides were recognized per protein and (b) the peptide length was required to be at least seven amino acids long. The false discovery rate (FDR) at the peptide and protein levels was set to 0.01. To filter out nonspecific proteins, a limma-based differential analysis was performed comparing empty vector control samples. Hit and candidate proteins were defined based on the following criteria. Candidate proteins were defined with an FDR ≤0.2 and an FC ≥1.5 and hit proteins were based on an FDR <0.05 and an FC >2. Statistically enriched hit and candidate proteins were analyzed using Panther pathway analysis software 14.0 (http://www.pantherdb.org; RRID:SCR_004869). The entire human proteome was used as a background reference. Statistical significance of the enrichment of each pathway was assessed using Fisher's exact test, with the Benjamini–Hochberg false discovery rate (FDR) correction for multiple comparisons (adjusted *p* value <0.05). Human Mitochondrial proteins were identified according to the Human Mitocarta 3.0 database (https://www.broadinstitute.org/files/shared/metabolism/mitocarta/human.mitocarta3.0.html; RRID:SCR_004869). The subcellular localization of the top 100 candidate proteins was assigned according to the GeneCards database version 5.6 (https://www.genecards.org;

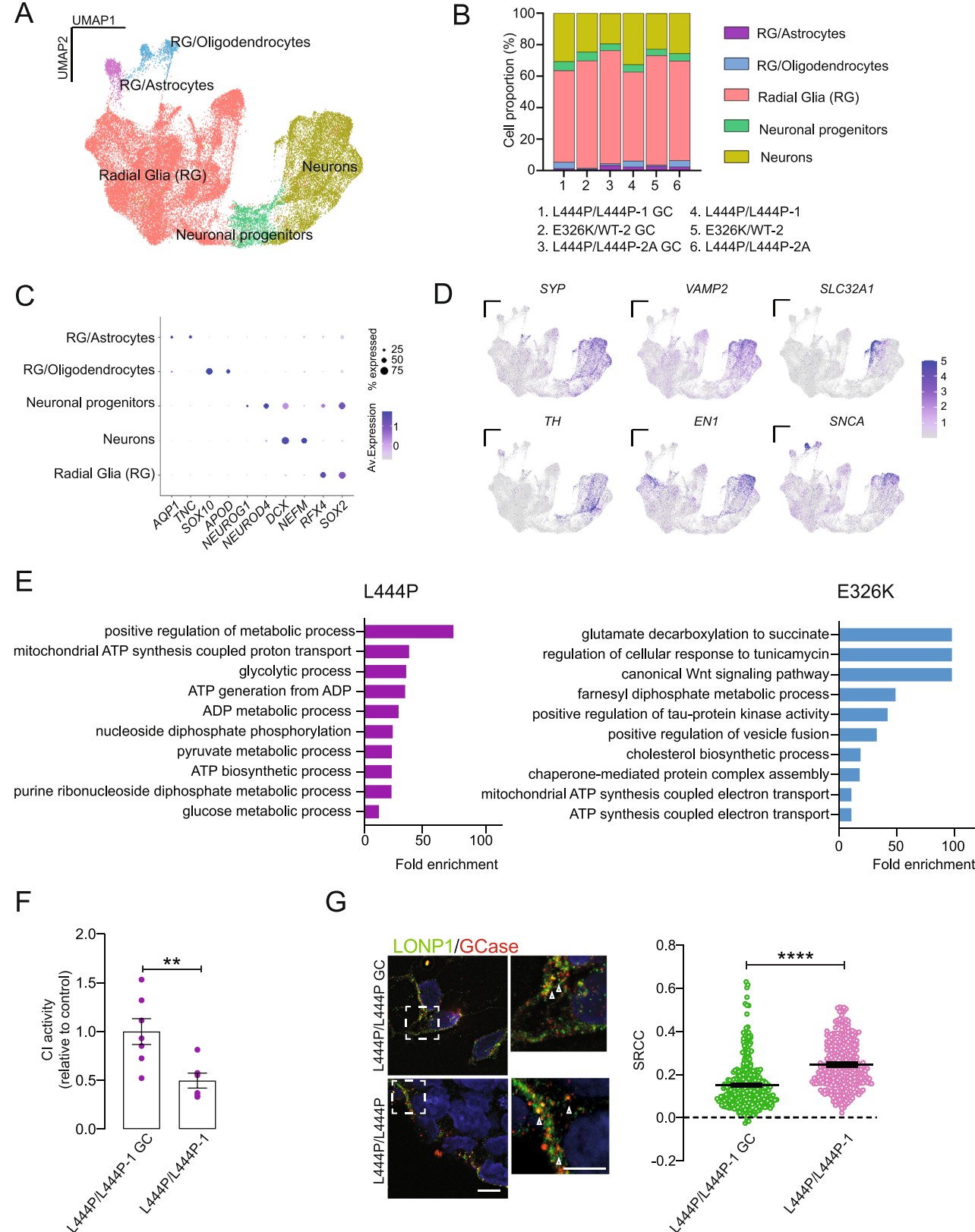

RRID:SCR_002773) by using a confidence level ≥4. The visualization of the protein subcellular localization was generated with Cytoscape version 3.9.0 (https://cytoscape.org; RRID:SCR_003032). The WT-GCase interactor list was used as the source node, fold-changes as the source attribute, and subcellular localization as the target node. The distance and distribution between the nodes were randomly assigned.

**iMTS-ls profile generation**

Internal N-terminal matrix-targeting-like signal (iMTS-ls) analysis was performed as described in ref. 15. Multiple truncated GCase protein sequences were generated by sequentially removing N-terminal amino acids. GCase suffix sequences were then subjected to TargetP prediction (TargetP 2.0) in standard FastA format.

**Fig. 6 | scRNA-seq of *GBA1* mutant midbrain organoids reveals cell-type-specific mitochondrial vulnerability. A** UMAP cluster-integration dimensionality plot showing the unsupervised clustering of *GBA1* mutant midbrain organoids and isogenic controls. **B** Representation of the frequency of major cell clusters in *GBA1* mutant midbrain organoids and isogenic controls. Cell clusters were determined by the expression of specific markers. **C** Dot plot showing the expression levels of the indicated cell-type-specific genes visualized in the UMAP plots. **D** Expression levels of the indicated synaptic genes visualized in the UMAP plots. **E** Gene ontology pathway analysis of the DEGs showing the top dysregulated BPs between *GBA1* and isogenic midbrain organoids in DA neuronal clusters. **F** CI activity was measured in mitochondrial extracts from L444P/L444P *GBA1* organoids and corresponding isogenic controls. Data were normalized to protein content and expressed relative to the isogenic control. Mean ± SEM; unpaired two-tailed *t*-test; **P = 0.0093; each dot represents an individual midbrain organoid; n = 7 from three independent differentiation onsets. **G** ExM images showing LONP1 (green) and GCase (red) in L444P-L444P midbrain organoids and corresponding isogenic controls. Cell nuclei were counterstained with DAPI (blue). Scale bar, 10 μm. Colocalization analysis is shown on the right. Mean ± SEM; unpaired two-tailed *t*-test; ****P < 0.0001; images were taken from three independent experiments. Source data are provided as a Source Data file.

The mTP scores obtained were plotted against the corresponding amino acid position.

## Split-GFP mitochondrial localization

For mitochondrial targeting of $GFP_{1-10}$ ($MTS$-$GFP_{1-10}$), the N-terminal matrix-targeting signal (MTS) of mitochondrial subunit VIII of cytochrome c oxidase (COX8A) was added at the N-terminus of the $GFP_{1-10}$ β-strand sequence, as described in ref. 48. The $MTS$-$GFP_{1-10}$ construct was obtained from Integrated DNA Technologies IDT and cloned into pcDNA™5/FRT/TO (Thermo Fisher Scientific, # V652020). The stable T-Rex HEK cell line was generated as described above. For GCase split-GFP, a 27-bp linker followed by the GFP β11 sequence[48] was added at the C-terminus of the GCase cDNA sequence. For MT-GCase split-GFP, we cloned a construct in which we inserted the mitochondrial targeting sequence of COX8A along with a nine-amino acid (aa)-long linker at the N-terminus and the 11th β-strand sequence of GFP at the C-terminus of GCase. Constructs were obtained from VectorBuilder. E326K or L444P mutation was inserted into the WT construct by site-directed mutagenesis using the QuickChange XL kit (Agilent) according to the manufacturer's protocol. For $mtGFP_{1-10}$ induction, $2 \times 10^4$ $mtGFP_{1-10}$ T-Rex HEK cells were seeded on Geltrex (Thermo Fisher)-coated chambered cell culture slides (Ibidi) for live-cell imaging or immunostaining and treated with 50 to 200 ng/ml doxycycline. GCase split-GFP constructs were transfected with Viafect (Promega) according to the manufacturer's protocol.

## Live-cell imaging

For live-cell imaging, cells were washed once with OptiMEM (Gibco) and then incubated for 30 min at 37 °C with 100 nM MitoTracker red CM-$H_2$Xros (Thermo Fisher Scientific) in OptiMEM. Cells were washed once with OptiMEM, in which the cells were kept for imaging. Images were acquired using a Leica TCS SP8 confocal microscope (Leica, Germany) equipped with a 100×/1.4 numerical aperture oil-immersion objective. Images were analyzed using Diffraction PSF 3D and DeconvolutionLab2 plugins in Fiji-ImageJ version 2.3.0/1.53q (https://fiji.sc; RRID:SCR_002285). For A-SYN PFF experiments, iPSC-derived neurons were treated with 0.25 μM Alexa Fluor 594-labeled PFFs (594-PFF) with or without cotreatment with 1 μM CDDO-Me (Cayman Chemical). After 24 h, the cells were incubated with 100 nM MitoTracker Green (Invitrogen, MA, USA) in a neuronal medium for 30 min at 37 °C. A CellBrite™ Steady 488 Membrane Staining Kit (Biotium) was used to visualize cell membranes according to the manufacturer's instructions. Images were acquired using a Leica TCS SP8 confocal microscope (Leica, Germany) with a 63×/1.4 numerical aperture oil-immersion objective. Z-stacks were acquired for calculation of the PFF particle area and fluorescence intensity. For each condition, 5-8 images were acquired from at least four independent experiments. Data were obtained from $n > 20$ cells per experiment per condition. For the quantification of colocalization and image processing, images were analyzed using the "Analyze particles" and "EzColocalization" plugins in Fiji-ImageJ.

## Generation of induced pluripotent stem cells and gene correction

Skin fibroblasts were reprogrammed by nucleofection with pCXLE-hOct3/4 (RRID:Addgene_27076), pCXLE-hSK (RRID:Addgene_27078), and pCXLE-hUL (RRID:Addgene_27080) plasmids[49] using the Amaxa nucleofection kit for human dermal fibroblasts (Lonza, VPD-100) and program P-022 of the Nucleofector 2b (Lonza). Nucleofected fibroblasts were plated in six-well plates coated with Matrigel (Corning) in DMEM supplemented with 10% FBS (Gibco) and 1% GlutaMAX Supplement (Gibco). The following day, the medium was changed to DMEM+/+ (DMEM with 10% FBS and 1% GlutaMAX Supplement and 1% Pen/Strep (Millipore)) supplemented with 2 ng/ml recombinant basic human fibroblast growth factor (FGF2, Peprotech). On day 3 or 4 post nucleofection, the medium was changed to E8 medium composed of DMEM F12 with HEPES (Gibco), 128 ng/ml ascorbic acid (Sigma–Aldrich), 1x insulin-transferrin-selenium (Thermo Fisher Scientific), 10 ng/mL FGF2 (Peprotech), 500 ng/ml heparin (Sigma–Aldrich), and 2 ng/ml TGFβ1 (Peprotech). E8 medium was supplemented with 100 μm sodium butyrate and 0.1% Pen/Strep. Colonies started to appear from day 14 onward. Induced pluripotent stem cells (iPSCs) were cultured on Vitronectin XF (StemCell Technologies) in E8 medium. Gene correction for the L444P mutation was performed as previously described in ref. 50. The crRNA for the gene correction of the E326K mutation was designed with an online CRISPR design tool according to ref. 51. The guides and ssODN sequences are listed in Supplementary Table 2. One hour before nucleofection, 10 μM Rock inhibitor was added to the iPSC medium. About 240 nM crRNA (IDT): Atto550-labeled tracrRNA (IDT) duplex was complexed with 124 μM Cas9 to form the ribonucleoprotein complex (RNP complex). iPSCs ($1.6 \times 10^6$) were nucleofected with the RNP complex and 16 μg of ssODN using 100 μl of Ingenio nucleofection solution (Mirus) with program B-016 of Nucleofector 2b. Following nucleofection, the cells were FACS sorted for Atto550-positive cells using a FACS Aria II with a 100-μm nozzle. After sorting, $1 \times 10^4$ cells were plated per 10-cm dish. Colonies were picked and sequenced by Sanger sequencing. The top five possible exonic off-target effects predicted by https://cctop.cos.uni-heidelberg.de:8043/ (RRID:SCR_016963) were checked by Sanger sequencing.

## Differentiation of iPSCs into neuronal precursor cells and dopaminergic neurons

The generation of neuronal precursor cells (NPCs) from the patient and isogenic control iPSCs as well as the differentiation from NPCs to dopaminergic neurons, was performed according to ref. 52. NPCs were cultured in N2/B27 medium, consisting of DMEM-Ham's F12 medium and neurobasal medium (1:1), 0.5% N2 (Gibco), 1% B27 (Gibco), 1% P/S, and 1% GlutaMAX supplemented with 3 μm CHIR99021 (Axon Medchem), 0.5 μM puromycin (Merck Millipore), and 150 μM ascorbic acid (AA, Sigma–Aldrich). NPCs were split into Matrigel-coated wells every 5–6 days at a ratio of 1:10. To start differentiation, $1.25 \times 10^6$ cells were split into a six-well plate, and the following day, the medium was changed to differentiation medium [N2/B27 medium supplemented with 100 ng/ml FGF8 (Peprotech), 1 μM phorbol 12-myristate 13-acetate

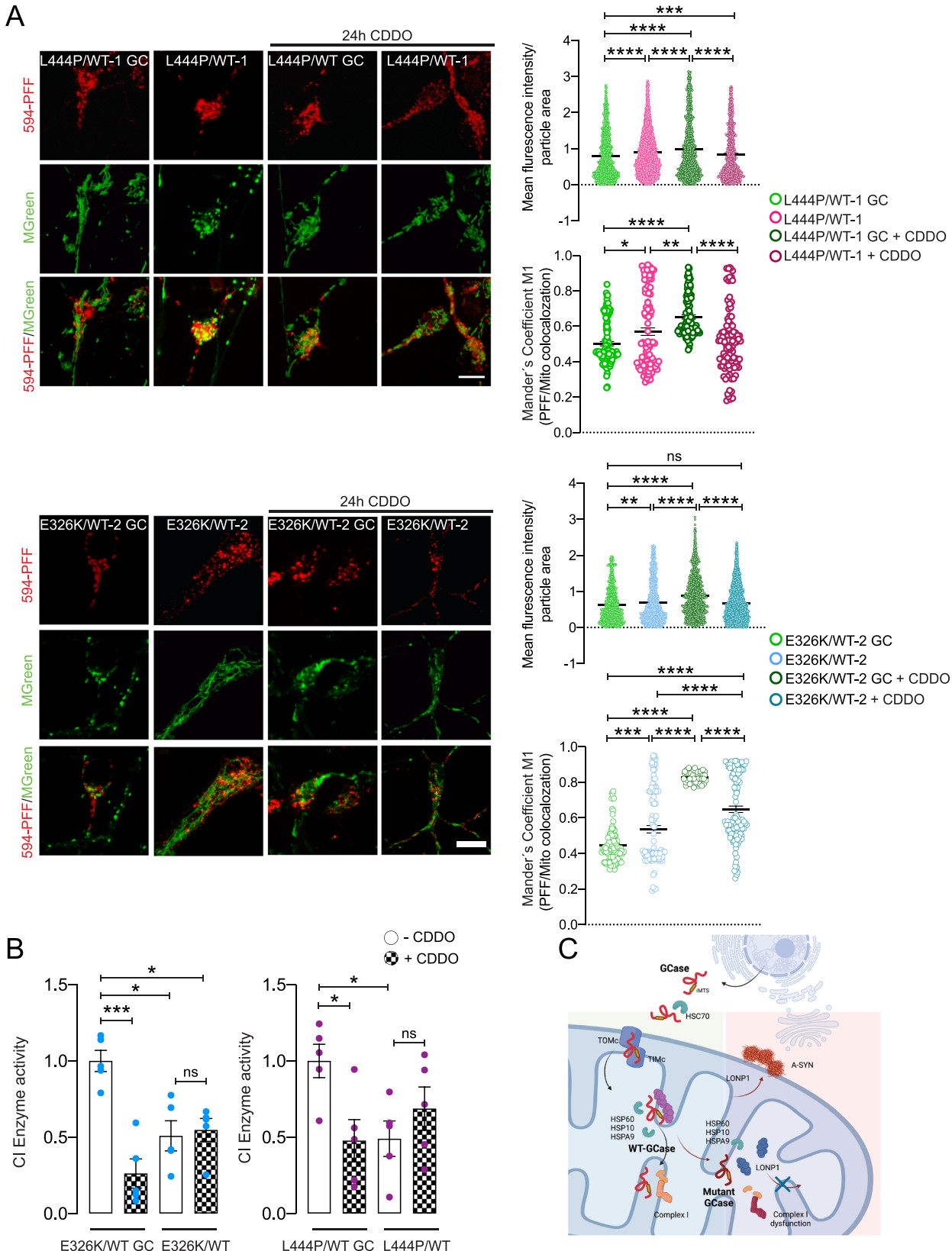

(PMA) and 200 μM AA]. On day 8, the medium was changed to maturation medium [N2/B27 medium supplemented with 10 ng/ml brain-derived neurotrophic factor (BDNF) (Peprotech), 10 ng/ml glial cell-derived neurotrophic factor (GDNF) (Peprotech), 1 ng/ml TGF-β3 (Peprotech), 0.5 mM dibutyryl cAMP (dbcAMP) (Applichem), and 200 μM AA] supplemented with 1 μM PMA for 2 days. From day 10

onward, the cells were cultured in a maturation medium without PMA. Maturation was reached after 14 days in the maturation medium.

**Western blot analysis**

Cells were collected and washed once with PBS. The pellets were lysed, and the protein concentration was determined using the Pierce BCA

**Fig. 7 | The mitochondrial LONP1 protease contributes to GCase-linked A-SYN accumulation. A** iPSC-derived neurons of the indicated genotypes were incubated with Alexa Fluor 594-labeled A-SYN preformed fibrils (PFFs) with or without CDDO-Me treatment. Live-cell imaging was performed 24 h after treatment to assess the content of internalized A-SYN PFFs. To label mitochondria, iPSC-derived neurons were incubated with MitoTracker Green. The PFF intracellular signal was calculated using ImageJ and expressed as the fluorescence intensity normalized to the particle area. A-SYN PFF intramitochondrial accumulation was assessed by measuring the colocalization between Alexa Fluor 594 and the MitoTracker Green signal. Mean ± SEM; two-way ANOVA with Bonferroni post hoc test; L444P/WT lines, upper panel: ****$P < 0.0001$, ***$P = 0.0004$; lower panel: **$P = 0.0023$, *$P = 0.0165$,

****$p < 0.0001$. E326K/WT lines, upper panel, **$P = 0.0054$, ****$P < 0.0001$; lower panel, ***$P = 0.0008$, ****$P < 0.0001$; for each condition, five to eight images were acquired from $n = 4$ independent experiments. Data were obtained from more than 20 cells per experiment per condition. Scale bar, 10 μm. **B** iPSC-derived neurons of the indicated genotypes were incubated with or without CDDO-Me treatment, and CI activity was measured in mitochondrial extracts. Mean ± SEM; two-way ANOVA with Bonferroni post hoc test; left panel, ***$p = 0.0004$, *$p = 0.019$, 0.021, in sequence; right panel; *$p = 0.0175$, 0.0148, in sequence; $n = 5$ independent experiments. **C** Graphical representation of GCase import and function in mitochondria and the proposed role of LONP1 in WT and mutant GCase models. Source data are provided as a Source Data file. Images were created with BioRender.com.

Protein Assay Kit (Thermo Fisher Scientific). Protein (20–30 μg) was mixed with 6x Laemmli containing 12.5% β-mercaptoethanol (Roth). Protein samples, 20–30 μl of eluate of the Flag precipitate or the eluate of the endogenous PD were loaded on self-casted 7.5–15% acrylamide gels (Applichem) or on precast NuPage 4–12% Bis-Tris Protein Gels (Thermo Fisher Scientific). Proteins were transferred to PVDF membranes. The PVDF membranes were blocked in TBS 0.1% Tween containing 5% milk (nonfat dried milk powder, PanReac AppliChem) or 5% bovine serum albumin (Serva). The primary antibody was diluted in 5% Roche Block solution (Roche) or 5% BSA in TBS-T supplemented with 0.04% sodium azide. Secondary antibodies were diluted in a blocking solution. The antibodies are listed in Supplementary Table 3.

### Soluble and insoluble A-SYN fractionation
Extraction and detection of Triton-soluble (T-sol) and Triton-insoluble (T-insol) alpha-synuclein was performed as described in ref. 53. Individual organoids were lysed in 1% Triton X-100 extraction buffer (1% Triton X-100, 150 mM NaCl, 10% glycerol, 25 mM HEPES pH 7.4, 1 mM EDTA, 1.5 mM MgCl2) supplemented with 1X PIC, 50 mM NaF, 2 mM NA$_3$VO$_4$, and 0.5 mM PMSF. Samples were homogenized with a pestle and incubated on a platform shaker in an ice-water slurry for 30 min, followed by three freeze/thaw cycles and ultracentrifugation at 100,000×$g$ at 4 °C for 30 min. The supernatant was removed and labeled T-sol. The remaining pellet was washed in Triton X-100 extraction buffer, followed by another ultracentrifugation at 100,000×$g$. The pellet was extracted in 2% SDS buffer containing 50 mM Tris, pH 7.4, and 1X PIC, boiled for 10 min at 100 °C, and labeled the T-insol fraction. T-insol fractions were sonicated in a cup horn probe sonicator (Qsonica – Q700) and boiled again for 10 min at 100 °C. The lysate was ultracentrifuged at 100,000 × $g$ for 30 min at 21 °C. The supernatant was labeled the SDS-soluble fraction. Protein concentrations were detected using a BCA assay and 30 μg of total protein in each condition were loaded.

### Blue native electrophoresis and in-gel activity assay
Mitochondrial pellets were isolated from NPCs and iPSC-derived neurons and processed for blue native electrophoresis (BN-PAGE) as previously described in refs. 54,55. Samples were solubilized with digitonin at a detergent-to-protein ratio of 4:1, and 40 μg of mitochondrial protein was loaded per lane in precast NativePAGE™ 3–12% Bis-Tris gels (Invitrogen™). After electrophoresis, proteins were transferred to PVDF membranes at 30 V overnight and probed with the appropriate antibody.

### Midbrain organoid generation
Midbrain-like organoids were generated by a protocol adapted from ref. 20. At day 0, iPSC colonies were dissociated into single cells using Accutase. In each well of a 96-well U-bottom low-attachment plate, $10^4$ iPSCs were seeded in a 1:1 mix of DMEM F12 and neurobasal medium supplemented with 0.5% N-2 supplement, 1% B27 supplement without vitamin A, 1% NEAA, 0.01% β-mercaptoethanol, 1 μg/ml heparin, 10 μM SB431542, 200 ng/ml LDN, 0.7 μM CHIR99021, and 50 μM Rock inhibitor. On day 4, the medium was changed to DMEM F12/neurobasal

(1:1) supplemented with 0.5% N$_2$ supplement, 1% B27 supplement, 1% NEAA, 0.1% β-mercaptoethanol, 1 μg/ml heparin, 10 μM SB431542, 200 ng/ml LDN, 100 ng/ml SHH-C25II, 1 μM purmorphamine, 7.5 μM CHIR99021, and 100 ng/ml FGF8. On day 7, embryoid bodies (EBs) were embedded in Matrigel using a protocol adapted from ref. 56. Matrigel was diluted in a 3:2 ratio with medium and used as an embedding mixture. EBs were washed in fresh medium and embedded in Matrigel in a six-well ultralow-attachment plate using the Matrigel-medium mixture. The Matrigel-EB mixture was incubated for 30 min at 37 °C, and on day 7, medium containing the following was added: neurobasal medium supplemented with 0.5% N2 supplement, 1% diluted B27 supplement, 1% GlutaMax, 1% NEAA, 0.1% β-mercaptoethanol, 2.5 μg/ml insulin, 200 ng/ml laminin, 7.5 μM CHIR99021, 100 ng/ml SHH-C25II, 1 μM purmorphamine, and 100 ng/ml FGF8. On day 8, the medium was changed to neurobasal medium supplemented with 0.5% N$_2$, 1% B27, 1% GlutaMax, 1% NEAA, 7.5 μM CHIR99021, 0.1% β-mercaptoethanol, 10 ng/ml BDNF, 10 ng/ml GDNF, 100 μM ascorbic acid, and 125 μM dbcAMP. On day 10, the medium was refreshed, and the concentration of CHIR99021 was reduced to 3 μM. On day 13, the concentration of CHIR99021 was further reduced to 0.7 μM until day 15. On day 20, the Matrigel was dissociated from the organoids. Organoid diameter measurement was performed at DIV 65 using the Analyze > Measure function, ImageJ.

### 10X Genomics scRNA sequencing
Midbrain organoids were dissociated into single-cell suspensions using a Worthington Papain Dissociation System kit (Worthington Biochemical), as previously described in ref. 57. Organoids (individual or pooled, 5/group, as indicated) were minced and incubated with 2.5 mL papain/DNAase solution on an orbital shaker for 30 min at 37 °C. Organoid suspensions were triturated using 1 ml low-attachment pipette tips and then incubated on an orbital shaker for 20 min at 37 °C. After enzyme inactivation, the cell suspensions were filtered with a 30-μm cell strainer and washed three times with HBSS. The final cell suspension was centrifuged at 300×$g$ for 7 min, and the cell pellet was resuspended in PBS-0.5% BSA. The cell concentration and viability were determined by microscopy using trypan blue staining (0.2%). scRNA-seq libraries were immediately generated using the 10X Chromium Next GEM Single Cell 3' Reagent Kit v3.1 (Cat. 1000128) according to the manufacturer's instructions and paired-end sequenced on an Illumina NovaSeq 6000 (SP Flow Cell) with a sequencing depth of 300 million reads per library. GEO accession number: GSE198033.

### Single-cell sequencing data analysis
The 10X Genomics pipeline cell ranger count was run to generate filtered gene-barcode matrices that were used as input for downstream analysis using the R package Seurat version 4.1.0 (RRID:SCR_007322) (Butler et al. 2018, Stuart et al. 2019). Low-quality cells were filtered out (detected genes >200, counts >500, and mitochondrial ratio <0.2), and genes with zero counts and genes that were expressed in fewer than ten cells were also removed. We analyzed 8037 cells and 8198 cells for the L444P/L444P-2 mutant and gene-corrected lines,

respectively, 6691 cells and 6223 cells for the E326K/WT-1 mutant and gene-corrected lines, respectively, and 6750 cells and 9886 cells for the L444P/L444P-1 mutant and gene-corrected lines, respectively. For clustering analysis, the dataset was SC transformed and normalized, and variation due to the cell cycle and mitochondrial expression was regressed out. For analysis at DIV 42, data from different 10x runs were integrated into Seurat using individual 10X runs as grouping variables (IntegrateData function). Principle components (PCs) were determined, and using the first 40 PCs, cells were clustered using a K-nearest neighbor (KNN) graph with a clustering resolution of 0.6, resulting in 21 clusters. Cell clusters were visualized using UMAP. In the UMAP projection, no clusters were found that could indicate cell doublets (doubletFinder_v3 function). Conserved markers were determined across all samples for each cluster (FindConservedMarkers function) using default settings (min.pct = 0.25, logfc.threshold = 0.25) and used for assigning cell-type to cell clusters. Clusters were manually assigned using La Manno et al as a reference [22]. For differentially expressed (DE) analysis, nonintegrated samples were analyzed using FindMarkers and MAST statistical tests ($p_{adj}$ >0,05, logFC >2). DE genes were analyzed using Panther pathway analysis software 14.0, and Biological Processes as a GO aspect (http://www.pantherdb.org; RRID:SCR_004869). Pseudotime analysis was performed in the L444P/L444P-2 gene-corrected line using Monocle 3 (version 1.0.0; https://cole-trapnell-lab.github.io/monocle3/; RRID:SCR_018685). Single cells isolated from five pooled organoids per time point were used for the analysis (DIV 42: 9692 cells, DIV 65: 8998 cells, DIV 100: 7422 cells). After data normalization and integration to reduce the batch effect, the Monocle 3 object was generated using the as.cell_data_set function, and cells were clustered using the cluster_cells function with a resolution of 1e-3. Clusters were manually assigned based on ref. 22. When ordering the cells along the trajectory using the "order_cells" function, SOX3-expressing cells were specified as the starting state. Nebulosa's kernel function was used to show the density estimation for cell-type specific markers in the UMAP plot. Scripts and R session info are available at https://doi.org/10.5281/zenodo.6475864[58]. The corresponding annotation file was downloaded from https://raw.githubusercontent.com/hbctraining/scRNA-seq/master/data/annotation.csv.

## Immunofluorescence and image analysis

Cells were washed once with PBS and fixed for 10 min with 4% paraformaldehyde (PFA, Sigma–Aldrich) at RT. After fixation, the cells were washed twice with PBS. The cells were blocked and permeabilized in PBS containing 0.1% Triton X (PBS-T) and 10% normal goat serum (NGS) for 1 h at RT. The primary antibody was diluted in PBS-T containing 5–10% NGS and incubated overnight at 4 °C. Secondary antibody (Invitrogen) was diluted in 5% NGS in PBS-T and incubated for 1 h at RT. The nuclei of the cells were stained by incubation with 1 μM 4′,6-diamidino-2-phenylindole dihydrochloride (DAPI, Invitrogen), and coverslips were mounted on glass slides with mounting medium (DAKO). Images were acquired with a 63 × 1.4 NA plan-apochromatic oil objective of a TCS SP8 confocal microscope (Leica Biosystems). Midbrain organoids were fixed in 4% (w/v) PFA for 1 h, and individual organoids were equilibrated in 30% sucrose in PBS overnight at 4 °C. The next day, the organoids were embedded in blocks in an optimal cutting temperature compound (OCT, Tissue-Tek). Slices with a thickness of 20 μm were cryosectioned and mounted on noncharged slides. Tissues were blocked in 10% (v/v) NGS in 0.5% Triton X-100 in PBS and incubated with primary antibodies overnight and secondary antibodies for 3 h. DAPI staining was performed at a concentration of 1 μM for 5 min, and after three PBS washes, tissues were mounted with DAKO mounting medium (DAKO) for image acquisition. The antibodies are listed in Supplementary Table 3. Whole-organoid quantification of TH + neurons was performed using the RapID Cell Counting tool[59] using the following parameters (max σ: 17; min σ: 4; overlap: 0.85; threshold: 0.04). At least three nonsequential cross-section tile

images per organoid (from different batches) were quantified. Whole-organoid confocal tile scan images were acquired using a Leica TCS SP8 confocal microscope (Leica, Germany) equipped with a 20×/1.4 numerical aperture oil-immersion objective. Tile-scanned fluorescence microscope images were stitched using the automated stitching function of the LasX software package. Cell nuclei were stained with DAPI, and total cells were detected using the blue-channel array (from an RGB image). Fluorescent cells were detected using red-channel or green-channel arrays.

## Neuromelanin staining (Fontana–Masson staining)-DAB staining

Human midbrain organoid sections were incubated in a fresh solution of 3:1 methanol (MeOH)/3% hydrogen peroxide at room temperature for 20 min. Slides were then washed in PBS 1 × 3 times for 5 min and blocked with NGS 10% in PBS + Triton X 0.2% for 1 h at room temperature. Primary antibodies were applied in NGS 5% in PBS + Triton X 0.2% solution overnight at 4 °C. Next, slides were washed in 1X PBS 3 times for 5 min, and secondary antibodies were applied to NGS 5% in PBS + Triton X 0.2% solution for 1 h at room temperature. ABC solution from Vectastain was prepared according to the manufacturer's instructions (VECTASTAIN® Elite® ABC-HRP Kit, Peroxidase (Standard) PK-6100) and applied to sections for 1 h at room temperature. Slides were then washed in 1X PBS three times for 5 min. DAB solution was prepared according to the manufacturer's instructions by diluting in 50 ml of 1x PBS with 50 μL of 3% H2O2. DAB solution was applied to the sections at room temperature for 30 s to 12 min, depending on when the visible reaction occurred. For the visualization of neuromelanin, the Fontana–Masson stain kit (Fontana–Masson Stain Kit; Sigma–Aldrich-HT200) was used. After DAB staining, slides were incubated in warmed ammonium silver solution at 58–60 °C for 30 min, according to the manufacturer's instructions. Slices were toned in Gold Chloride Solution for 30 s and then placed in Sodium Thiosulfate Solution for 1–2 min. Slides were eventually mounted with synthetic resin.

## Expansion microscopy

Expansion microscopy (ExM) was performed as described in ref. 60 with some modifications. Cells were blocked with 10% (v/v) normal goat serum (NGS) in 0.1% (v/v) Triton X-100 in PBS and incubated with primary antibodies in a blocking solution overnight. After a 3-h incubation with the corresponding secondary antibody (Alexa Fluor, Invitrogen), the samples were washed and treated with 0.1 mg/ml Acryloyl-X SE solution (Thermo Scientific) in PBS for 3 h at room temperature. The freshly prepared gelling solution consisted of Stock X solution (8.6% (w/v) sodium acrylate 33% (w/v), 2.5% (w/v) acrylamide 50% (w/v), 0.15% (w/v) N,N´-methylenebisacrylamide 2% (w/v), 11.7% (w/v) NaCl 5 M, and PBS 1X), water, 10% (w/v) TEMED and 10% (w/v) APS stock solution in a 47:1:1:1 ratio. Gel digestion was performed overnight in digestion buffer (0.5% (w/v) Triton X-100, 0.2% (v/v) EDTA 0.5 M, pH 8, 5% (v/v) Tris-Cl 1 M, pH 8, 4.67% (w/v) NaCl, and 8 U/ml proteinase K). The gelling solution was added to each well and covered by a 15-mm coverslip to ensure the formation of a smooth, flat, and thin gel. Coverslips were then incubated for 1 h at 37 °C for complete polymerization. The gel was expanded in water for 1 h and mounted in 10 μg/mL poly-L-ornithine-coated coverslips to immobilize the gel for picture acquisition. Midbrain organoids were fixed, and immunofluorescence staining was performed as described above. Sections were treated with 0.1 mg/ml acryloyl-X SE solution in PBS at room temperature overnight. Gelation was performed in a 47:1:1:1 ratio of Stock X, 10% (w/v) TEMED, 10% (w/v) APS, and 0.5% (w/v) 4-hydroxy-TEMPO stock solutions. Gel digestion and expansion were performed as described above. Images were acquired using a Leica TCS SP8 confocal microscope (Leica, Germany) equipped with a 100×/1.4 numerical aperture oil-immersion objective. For each condition, five images were acquired from at least three independent experiments. Images were analyzed using

Diffraction PSF 3D, DeconvolutionLab2, and EzColocalization plugins in Fiji-ImageJ. GraphPad Prism version 9.0.0 (RRID:SCR_002798) was used to calculate Spearman's rank correlation value (ρ) to identify the colocalization of fluorescence signals. The antibodies are listed in Supplementary Table 3.

## Mitochondrial isolation

Mitochondria were isolated from $10 \times 10^6$ cells or individual organoids using the Qproteome Mitochondrial isolation kit (Qiagen) with slight alterations to the manufacturer's protocol. Cells or individual organoids were harvested, washed in 0.9% NaCl, and incubated for 10 min at 4 °C in a lysis buffer. The homogenate was centrifuged at $1000 \times g$ for 10 min at 4 °C, and the supernatant was designated the cytosolic fraction. The pellet was resuspended in a disruption buffer and mechanically passed through a 26-gauge needle ten times. The enriched nuclear fraction was pelleted by centrifugation at $1000 \times g$ for 10 min and homogenized in a disruption buffer. Next, the supernatant was centrifuged at $14,000 \times g$ for 30 min at 4 °C and resuspended in mitochondria storage buffer (enriched mitochondrial fraction) or respirometry assay buffer. All buffers (except the mitochondria storage buffer) were supplemented with 1:100 protease inhibitors (provided with the kit). Where indicated, cells were treated with 20 ng/µl digitonin (Sigma–Aldrich) or 0.1% Triton X-100 in PBS. For the proteinase K protection assay in T-Rex cells, freshly isolated mitochondria were resuspended in mitochondria storage buffer and treated with 2 mg/ml proteinase K (New England Biolabs GmbH) for 30 min on ice to digest surface-exposed proteins. The reaction was stopped by adding 2 mM PMSF Protease Inhibitor (Thermo Fisher). For mitochondrial isolation in NPCs, mitochondria treated with digitonin and Triton X-100 were pelleted down at $14,000 \times g$ for 30 min, lysosomal proteins in the supernatant were discarded, and the pellet was resuspended in 20 µl of mitochondria storage buffer for immunoblotting analysis. Twenty micrograms of total fraction, 10 µg of mitochondrial fraction, and a total volume of detergent-treated mitochondrial fraction were loaded.

## A-SYN PFFs

The production and purification of recombinant human A-SYN was conducted according to ref. 61. A-SYN cDNAs were cloned into pET-21d(+) DNA (Novagen, Merck Millipore, #69743), and the plasmids were expressed in BL21DE3 *E. coli* (Novagen, Merck Millipore, #69450; RRID:CVCL_M639). Cultures of 750 ml were grown to mid-log phase, and isopropyl-1-thio-3-D-galactopyranoside was added to 0.4 mM. After 2 h, the cells were pelleted, washed in PBS, and resuspended in 50 ml of 20 mM HEPES, 100 mM KCl, pH 7.2. The resuspended bacteria were heated to 90 °C for 5 min. Aggregated protein was removed by centrifugation (40 min at $40,000 \times g$ at 4 °C). Contaminating nucleic acids and proteins were removed by ion exchange chromatography on Q-Sepharose Hi-Trap columns equilibrated with Solution A (50 mM Tris pH 7.4; Amersham Biosciences). Purified A-SYN was eluted by loading the soluble fraction and applying an increasing gradient of Solution B (50 mM Tris, pH 7.4, 1 M KCl). SYN-containing fractions were pooled and chromatographed on a Superose 12 column (Amersham Biosciences) in 20 mM HEPES, pH 7.4, 100 mm KCl. The monomer was aliquoted and frozen at −80 °C. For the preparation of PFFs, the A-SYN monomer was shaken at 1000 rpm for 7 days. PFFs were validated by electron microscopy, a sedimentation assay at $100,000 \times g$ for 60 min, the thioflavin T assay, and primary cortical neurons by p-Ser129-A-SYN immunostaining. Preformed fibrils were labeled using the Alexa Fluor 594 kit (Thermo Fisher, #A10239) according to the manufacturer's instructions. For the treatment of neurons and organoids, A-SYN PFFs, which were generated at a concentration of 5 mg/mL, were vortexed and diluted with Dulbecco's phosphate-buffered saline to 100 µg/mL and then sonicated (10-s pulses with 30% amplitude six times every 2 min) using an HTU SONI-

130 sonicator (G. Heinemann, Germany). A-SYN PFFs were then diluted in neuronal or organoid media and added to cultures.

## Virus production

Verified Mission lentiviral plasmids encoding nontargeting shRNA (#SHC016) and HSPA8/HSC70 shRNA (#TRCN0000017279, #TRCN0000017281) in the pLKO.1-puro vector backbone were purchased from Sigma–Aldrich (RRID:Addgene_8453). The sequences of shRNA-TIM23 used by ref. 62 were cloned into pLV(shRNA)-Puro-U6 (purchased from VectorBuilder, VB210128-1106hrg and VB210129-1039mum); pLV[shRNA]-Puro-U6 > Scramble-shRNA (purchased from VectorBuilder, VB010000-0005mme) was used as a nontargeting shRNA. For lentiviral production, HEK cells were transfected with the plasmids expressing the shRNAs together with the lentiviral packaging plasmid psPAX2 (RRID:Addgene_12260) and the envelope plasmid pMD2.G (RRID:Addgene_12259) using TransIT-X2 (Mirus). In short, HEK 293 T cells were seeded in a 10-cm dish to reach 80% confluency the day after transfection. The next day, the medium was changed. On days 4 and 5 after transfection, the medium was collected and filtered through a 0.45-µm PVDF membrane. To concentrate the virus, the filtered supernatant was centrifuged in a Vivaspin column (Sartorius Stedim Biotech) at 3000 RCF at 4 °C until the volume reached 500–1000 µl. The virus-containing supernatant was collected, and the concentration of p24 particles was determined with Lenti-X GoStix Plus (Takara). Cells were infected with equal concentrations of p24 particles for the scramble and respective shRNA.

## GCase activity assay

Cells or midbrain organoids were lysed by sonication in $H_2O$ containing 0.01% Triton, and the protein concentration was determined by BCA. For enzymatic assay measurement, 10–20 µg of protein was incubated for 30 min at room temperature with 25 µl of McIlvaine buffer 4X (0.4 M citric acid, 0.8 M Na2HPO4), pH 5.2, AMP-DNM (*N*-(5-adamantane-1-yl-methoxy-pentyl)-deoxynojirimycin) at a final concentration of 5 nM, and H2O to a final volume of 100 µl. 4-Methylumbelliferyl-β-D-glucopyranoside (MUB-Glc; Glycosynth, Warrington, UK) was dissolved in 200 mM sodium citrate phosphate buffer by heating to 60 °C. At the end of incubation, 25 µl of 4-MU was added at a final concentration of 6 mM and incubated for 2 h at 37 °C. As a control, 1 mM condurbitol B epoxide (CBE, Calbiochem) was used. The fluorescence was recorded after transferring 20 µl of the reaction mixture to a microplate and adding 180 µl of 0.25 M glycine, pH 10.2. Data were calculated as picomoles of converted substrate per milligram of cell protein per hour.

## A-SYN ELISA

The levels of total A-SYN were measured in midbrain organoid homogenates using the hSYN total ELISA kit (847-0108000103, Roboscreen Diagnostic, Leipzig, Germany), according to the manufacturer's instructions. The optical density was read at 450 nm on a microplate reader (Bio-Rad). Data were normalized to the protein content.

## Mitochondrial respiration

Mitochondrial function was assessed in live cells using an XFp and XFe96 Extracellular Flux Analyzer (Agilent Technologies). A total of $1 \times 10^4$ T-Rex HEK or $1.5 \times 10^5$ iPSC-derived neurons were seeded in XFpSeahorse microplates and allowed to adhere overnight (for HEK cells) or for 7 days (iPSC-derived neurons). Measurements of the oxygen consumption rate (OCR) were performed in a freshly prepared assay medium, pH 7.4 (Seahorse XF DMEM), according to the manufacturer's protocol. The OCR was measured before and after the serial addition of 20 or 1 µM oligomycin, 1 or 5 µM carbonyl cyanide p-trifluoromethoxyphenylhydrazone (CCCP), 2 or 1 µM antimycin A and 2 or 1 µM rotenone to T-Rex HEK cells or iPSC-derived neurons,

respectively (all from Sigma–Aldrich). Following each injection, three measurements for a total period of 15 min were recorded. The data were analyzed using Wave 2.6 software (RRID:SCR_014526), and OCR parameters (basal respiration, maximal respiratory capacity, respiratory reserve, and ATP-linked respiration) were calculated. At least three technical replicates per condition were used, and the experimental values were normalized to the protein content per well via a BCA assay.

## Mitochondrial complex activity assays

Mitochondria were isolated from HEK cells, iPSC-derived neurons, or midbrain organoids using the Qproteome Mitochondrial isolation kit as described above. Complex I (NADH oxidase/coenzyme Q reductase) was measured using the MitoCheck Complex I Activity Assay kit (Cayman Chemical, cat# 700930). The rate of NADH oxidation, which is proportional to CI activity, was determined by a decrease in absorbance at 340 nm over 15 min in the presence of ubiquinone and potassium cyanide to inhibit complex IV and prevent oxidation of ubiquinone. To assess CI, CII, and CIV function, we used a respirometry approach based on XFp Extracellular Flux Analysis[63]. To this end, 3 mg of purified fresh mitochondria were resuspended in 200 μl of MAS buffer (70 mM sucrose, 220 mM mannitol, 5 mM $KH_2PO_4$, 5 mM $MgCl_2$, 1 mM EGTA, 2 mM HEPES pH 7,4) and seeded in XFpSeahorse microplates. The plate was centrifuged at $2000 \times g$ for 5 min at 4 °C. The OCR was measured before and after the serial addition of pyruvate + malate (5 mM each) + ADP 3,5 mM or 1 mM Succinate + 4 μM rotenone, 4 μM rotenone + 8 μM antimycin A, 0,5 mM TMPD (N,N,N′,N′-tetramethyl-p-phenylenediamine dihydrochloride, Santa Cruz Biotechnology) + 1 mM ascorbic acid, and 50 mM azide. Following each injection, three measurements for a total period of 15 min were recorded. Complex I-, II-, and IV-dependent respiration was calculated by subtracting OCR values from the substrates (Pyruvate + malate + ADP for CI, Succinate + rotenone for CII, and TMPD + ascorbic acid for CIV) subtracted from the ones from the inhibitors (rotenone for CI, antimycin A + rotenone for CII and azide for CIV). The experimental values were normalized to the protein content per well via a BCA assay.

## Statistics and reproducibility

Statistical differences among groups were evaluated using GraphPad Prism Version 9.3.1. All experiments were performed at least in triplicate, and quantitative data are presented as the mean ± SEM. Sample sizes were chosen based on previous studies. All samples were included in the analysis. Blinding was not performed in this study. Differences among groups were assessed using an unpaired two-tailed Student's t-test or ANOVA multiple comparisons, as indicated in the figure legends.

## Reporting summary

Further information on research design is available in the Nature Portfolio Reporting Summary linked to this article.

## Data availability

A reporting summary for this article is available as a Supplementary Information file. The main data supporting the findings of this study are available within the article and its Supplementary files. The mass spectrometry proteomics data have been deposited in ProteomeXchange via the PRIDE partner repository under the accession code PXD032155. For scRNA-seq data, GEO accession number GSE198033. Protocols have been deposited at https://doi.org/10. 17504/protocols.io.x54v9d44mg3e/v1. Scripts, packages, and R session info used in this study are available at https://doi.org/10.5281/ zenodo.6475864. Publicly available databases were used in this manuscript, including the UniProt Homo sapiens database

(UP000005640), Human Mitocarta 3.0 database (https://www. broadinstitute.org/files/shared/metabolism/mitocarta/human. mitocarta3.0.html; RRID:SCR_004869), and GeneCards database version 5.6 (https://www.genecards.org; RRID:SCR_002773). Further information and request for raw data can be provided upon request to Dr. Michela Deleidi. Source data are provided with this paper.

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

## Acknowledgements

The authors thank Johannes M Herrmann (University of Kaiserslautern), Peng Qui (Georgia Institute of Technology), and Frank Stein (Proteomics Core Facility, EMBL) for helpful discussions. We acknowledge the

funding support of the Helmholtz Association Young Investigator Award (VH-NG-1123, to M.D.), Juniorprofessuren-Programm BW (Az:7365.521/ (16), to M.D.), EU Joint Program—Neurodegenerative Disease Research (JPND) project (GBA-PaCTS, 01ED2005B, to M.D. and A.H.V.S.), Instituto de Salud Carlos III-MINECO/European FEDER Funds Grant PI20-00057 (to C.U.), Comunidad Autónoma de Madrid/ERDF-ESF P2018/BAA-4403 (to C.U.), DZNE (PD research; MIGAP study, to K.B. and T.G.), and EMBO ALTF-1013-2019 (to M.J.P.) for this project. This research was funded by Aligning Science Across Parkinson's [Grant number: ASAP-000420] through the Michael J. Fox Foundation for Parkinson's Research (MJFF) (to M.D. and A.H.V.S.).

## Author contributions

Conceptualization, P.B., M.J.P., and M.D.; formal analysis, P.B., M.J.P., H.R., F.B., S.K., F.S., G.C., C.U., and M.D.; investigation, P.B., M.J.P., S.K., H.R., M.I., F.S., C.G., F.B., M.O., H.H., A.M.C., C.U., and M.D.; writing—original draft, P.B. and M.D.; writing—review and editing, P.B., M.J.P., H.R., F.B., M.O., A.H.V.S., C.U., and M.D together with all other co-authors; funding acquisition, C.U., A.H.V.S., and M.D.; supervision, M.D.

## Competing interests

The authors declare no competing interests.
