## [Peer Review File · Nature Communications]

REVIEWER COMMENTS

Reviewer #1 (Remarks to the Author):

The data describes a novel potential role and localization of GCCase in mitochondria, and suggest that GCCase mutations may directly affect mitochondria function. Overall the data are interesting and highly novel, providing potential explanations for how GCCase mutants may cause neuronal dysfunction. However, there are several issues that reduce enthusiasm for publication. Most of the data showing the localization of GCCase in mitochondrial rely on artificial overexpression models of tagged proteins in HEK cells. More in-depth studies should be done on the endogenously expressed protein to rule out the possibility that mito localization doesn't occur from dramatic overexpression. Critical controls were missing in many assays. For example specificity was not addressed, through expressing another tagged hydrolase at similar levels as GCCase. The mitochondrial functional studies are well performed overall, however there is no evidence to suggest that the effects occur directly from GCCase in mitochondria as opposed to indirect effects from lysosomal or mitophagy dysfunction. There are also some conceptual hurdles that should be addressed better. One is the fact that Gaucher disease is an established lysosomal but not mitochondrial disorder, and the study doesn't provide the percent of GCCase distribution within lysosomal and mitochondrial, or how this changes with mutations. The idea that mitochondria represent a quality control mechanism for mutant GCCase clearance is interesting, however if true, it should be very difficult to visualize mutant GCCase interacting with LONP1 without protease inhibitors, since the mutant would be rapidly degraded. Finally, the CDDO and PFF experiments do not add much to the study due to specificity issues and artificial nature of PFFs. If these issues could be addressed, the work would represent an important advance in the field. These are the major issues, and below are included some additional suggestions, which I hope the authors find constructive and help to improve the work.

1. Please provide citation for the statement "The p.E326K variant displays a limited impact on GCCase activity and a lower penetrance compared to the severe mutation p.L444P" line 93.
2. Please indicate whether the GCCase tag is on the N or C terminus of the protein and if the artificial tagged construct influences the normal processing of the enzyme, including glycosylation, cleavage, and transport to the lysosome. The blot in Fig 1B appears clean, however one would expect to see multiple bands corresponding to different glycosylated forms of GCCase at steady state.
3. I would suggest including the densitometry and activity quantification from S1A, B, into the main figure as these are important data. Baseline activity from empty vector or minus Dox condition. Does the expression of L444P increase the activity over the base line minus Dox?

4. An important control would include the overexpression of another FLAG-tagged lysosomal hydrolase at the same dramatically high levels as FLAG-GCase, to determine the specificity of the mitochondrial localization.

5. In figure S3, the staining appears to have worked well, however it is difficult to see the colocalization of red and green channels. They appear non-overlapping in most of the images.

5. In Figure 2E, the dMTS-GCase is critical in supporting the idea that GCase can traffic to mitochondria, however this figure should also include evidence that the dMTS mutant is stable and expressed at similar levels as compared to the MTS-versions.

6. Quantification of the split-GCase mitochondrial localization is required to determine if wt and mutant GCase differ. They look relatively equal however one would expect the mutants to be slightly lower owing to their instability.

7. The figure legend of S4 or text should define what each line means – what is “GC”?

8. Showing that the endogenously expressed GCase localizes with mitochondria is an important part of the study. Figure S4D and E are therefore critical and I would suggest to include in the main figure, however the quality of the blots should be improved. Endogenous GCase is difficult to pull down with commercially available antibodies. It is difficult to imagine such an efficient pull-down of unstable L444P GCase (lane 1) from the barely detectable input shown. Full blot should be shown here. Perhaps blotting for a known GCase interactor, such as LIMP2 as a positive control, would help, or colocalization with another mitochondrial marker to validate the findings. Quantification or some indication of this result is reproducible is also required.

9. Clarification of the endogenous co-IP methods are required in the methods section, including the amount of lysate used, define “PD” line 497, and the GCase antibody used for endogenous pull-down.

10. The immunofluorescence data in Figure S4G requires improvement and controls. For example, testing the antibody in a GCase knock-out is required. Please indicate what percentage of GCase is mitochondrial vs. lysosomal by adding in a colocalization study with LAMP1/ lysotracker etc.

11. Line 267, it is not clear how the data indicate a role for GCase in CI assembly, since CI assembly was not assessed in this assay.

12. Figure 4B, there appears to be some fragment of a blot image over top of the green plot in the graph.

13. Figure 4D requires more replicates and statistical analysis. Here, it would be critical to include a control such as general lysosomal inhibitor, to determine if the effect is specific for GCase path.

14. The decrease in mitochondrial function by GCase mutants is interesting, however mitochondrial deficits have been previously demonstrated by other groups through other mechanisms. For example, it isn't clear how the authors can conclude that this effect doesn't occur from a decline in mitophagy. Perhaps comparing to the dMTS-GCase mutant would help here.

15. In Figure 5C, D, is there a feasible way to quantify the organoid diameter, so that we can get an idea of the distribution of sizes in the culture well? Size variability might play an important role in functional phenotypes, eg due to dead centers that are deprived of media.

16. Overall, the baseline characterization of the organoids appears very good, however better quantifications of TH+ and/or MAP2+ should be done. Figure S6A-E, and G all require quantification. Figure 5B-D, and F requires quantification.

17. In figure S6C, D, more studies and quantification are required to confirm that these structures are actually neuromelanin occurring from TH+ midbrain neurons.

18. In organoid studies of Figure 5 and S6, there should be some indication that the data is reproducible between distinct organoid batches, not only between individual organoids. If different batches were used, this should be indicated in the figure legends.

19. There are panel label errors in Figure 5.

20. Measurements of neurodegeneration should be done in the organoids at DIV42, to determine if the readouts at this time point are confounded by non-specific cell death processes.

21. In Figure S8C, it is difficult to see 'high levels of GAD1 and TH' line 343. Perhaps a quantitative assay such as ELISA or western blot would help here.

22. Figure S10- please indicate which genes are measured here in the figure legend.

23. CDDO inhibits multiple pathways and the studies here do not add much to the overall study. It is also not clear why PFFs are needed since they induce artificial aggregation through mechanisms that are distinct from GBA1 mutations. These studies introduce some confusion to the work.

24. Ultrastructural studies are required to confirm that PFFs in figure 7A are intramitochondrial. Better resolution fluorescent images are also required, as it appears that large red colored puncta overwhelm the cytoplasm, which may produce artificial colocalization.

Reviewer #2 (Remarks to the Author):

In the manuscript "Glucocerebrosidase, a Parkinson's disease-associated protein, is imported into mitochondria and regulates complex I assembly and function", the authors aim to understand the role of Glucocerebrosidase (GCase) in mitochondria and its involvement in cellular energy metabolism through maintenance of respiratory chain complex I (CI) stability and integrity. The authors used wild-type and mutant GCase HEK 293 stable cell lines to show, mostly via a proteomic approach, the association between GCase and mitochondria and the interaction between mutant GCase and mitochondrial quality control proteins. They also used midbrain organoids differentiated from iPSCs carrying GBA1 mutations and an isogenic control to investigate the role of GCase in more relevant physiological conditions. The authors show the selective dysregulation of neuronal mitochondrial complex I (CI) in mutant DA neurons and speculate that this molecular dysfunction may be a key pathogenetic driver of Parkinson's disease.

The major concepts of the manuscript are novel and could contribute to providing new insights into neurodegenerative diseases, and particularly into PD pathogenesis. The manuscript is well written and, overall, the conclusions are well supported. The efforts of the authors to study GCase in human neurons during differentiation from iPSCs is certainly commendable. However, a large part of the experiments are performed in HEK lines by protein overexpression. To make the manuscript suitable for publication in a top-tier journal such as Nature Communications, the authors should provide

additional experiments to consolidate their findings regarding the role of Gcase in human iPSC-derived neurons.

Major comments:

1) The manuscript relies on single cell data to obtain insights into cell-type specific mechanisms, but crucial details of the methods for these analyses are missing. For example, what cutoffs did the authors use to define a DE gene? Was the Harmony integration run based on time points or cell lines used? How did the data look before integration? How was the resolution established when defining clusters? How was the pseudotime analysis performed? How can claims about trajectories be made? Did the authors detect any cluster(s) that might indicate cell doublets? Violin or box plots illustrating the distribution of the number of expressed genes, UMI, and mitochondrial percentages should be shown.

2) It is unclear from the figures and figure legends how many experimental batches or organoids were used. The statistical basis is also unclear in this manuscript: the authors often state “from three independent experiments”, but it is not obvious what this means. Do the authors present data from three separate organoids? Or did they pool data from several organoids generated on separate occasions? If the latter, from how many organoids? How many organoids gave rise to the findings shown in the main scRNA-seq figures? Data should not be reported in aggregate but in terms of individual organoids in order to experimentally substantiate the claims made in this study. The authors should also report the number of cells analyzed in the Results section.

3) TH expression at single cell resolution is mainly detected at day 42 DIV (as shown in Fig. 6 D, Supplementary Fig. 8C). What happens to TH at later time points?

It is unclear why the authors adapted the 2D differentiation protocol from Kim TW et al to generate a new and as yet unvalidated 3D midbrain differentiation protocol. It's not trivial develop a 3D protocol also starting from 2D, considering the high number of patterning factors used and the complexity of driving DA neuron differentiation. The authors should better characterize the generated midbrain organoids by showing the expression of postmitotic dopaminergic markers (e.g., Calb/GIRK2/DAT/ALDH1a) at protein level. Also, how does this protocol compare to other midbrain organoid protocols? The authors should project their single cell datasets onto available human midbrain organoid datasets, e.g., the recently published single cell dataset generated from human midbrain organoids by Fiorenzano et al (Nat Comm 2021).

4) Interestingly, the authors observe A-SYN accumulation at day 45 when DA neurons are not yet terminally differentiated. It would be beneficial for the manuscript to include an A-SYN immunofluorescence analysis with relative quantification at a later time point during midbrain organoid differentiation.

Additionally, the immunofluorescence presented in Fig. 5G does not seem to reflect the total A-SYN levels assessed by ELISA. The authors should comment on this point.

5) Accumulation of phosphorylated A-SYN is mainly observed in neurons in the form of Lewy bodies. Did the authors try to analyze phosphorylated A-SYN at day 45? I would strongly suggest that the authors analyze the accumulation of A-SYN and its phosphorylated form at later stages of differentiation, when neuromelanin is released and mature DA neurons are better modeled in a dish.

Minor comments:

1) Any residual floor plate (FP) progenitor cluster in Fig. 6A,B? Expression analysis at single cell level of the main markers of FP (FOXA2, LMX1A, LMX1B, CORIN, etc.) would be extremely useful.

2) In light of changes in A-SYN levels, the authors should show the expression of SNCA in Fig. 6 D not in aggregate but by reporting isogenic control and mutant iPSC lines as individual samples.

3) The authors cannot conclude from the analyses in Fig. 5E that “scRNAseq revealed a profound dysregulation of metabolic pathways” (page 10, lines 345–350).

Gene ontology pathway analysis only reveals an enrichment in metabolic pathways relating to mitochondrial ATP-related functions. The authors should change the wording in both the Results and Discussion sections.

Reviewer #3 (Remarks to the Author):

In this manuscript, Baden et al, perform bulk proteomics, single-cell approaches, and IHC analysis in cell lines and iPSC cell and organoid models to investigate GCase-related neurodegeneration. The major findings are that a portion of GCase can enter mitochondria via the TOM import machinery. GCase promotes assembly of CI and interacts with HSP60 / LONP1. Notably, disease associated mutations appear to impair these functions. The manuscript does highlight a number of powerful modern techniques which are a strength. Overall, the Reviewer viewed this manuscript with some positivity, but a few major and a collection of minor concerns need to be addressed before considering for publication. The clarity of the writing needs to also be improved.

Major concerns

1) If I am not mistaken, all biochemical experiments are based on overexpression of GCase and mutants, therefore it is a bit difficult to determine whether this is physiologically relevant or an artifact. I realize this is not an easy limitation to address. Figure 4C seems to address this a bit but this finding needs to be expanded on.

2) Many of the experiments are either not controlled properly, or the controls need a better explanation in the text. i.e I am not sure what is the Ref library for all of the GO analyses, or what is the "CT" for Co-IP experiments. If CT is untransfected cell lysates, then that's a good control for the IP experiments but still ab it worrisome for the data presented in Figure 2.

3) Every protein tested in the Co-IP experiments came down with the FLAG-tag, thus the authors really need to show at least one negative control for the WBs that does not come down with Flag IP.

Minor concerns

Are there blots showing endogenous GCase in mitochondria? Fig 2A – weak band for “CT” .. ? that must be endogenous.

TMT analysis is not MS3-based and MS2 is just OK.

Line 131 should be isobaric, not isotopic

1C – not sure what the control (for relative enrichment) is here. Input? WT pulldown (no flag expression)?

1D – it isn't specified what is the reference of the GO analysis. All proteins in proteome? If so is that fair?

Association of the GCase with mitochondrial proteins is singled out while interactome with other organelles is ignored – other than previous literature is there a reason why lysosome or ER interactomes are ignored? Top 12 BP terms on the GO plot are NOT mitochondrial.

1E and S1E – again, not clear what is the control for this experiment. This data would be strengthened by an addition of negative i.e. protein that is not CO-IP'd with FLAG-tag, which would show specificity. It does look like EVERY protein tested comes down with the FLAG-tag, therefore showing a couple of proteins that do not is important.

Fig 2C-E – clever very nice!

206: “GCase is imported into mitochondria by the TOM protein import machinery”

- TOM import machinery consists of tom70, tom22, tom20, tom40, tom5, tom6 and tom7. HSP70 and Tim23 are not a part of the “TOM protein import machinery” - this needs to be cleared up the text
- This figure also lacks controls i.e. Co-IP of a protein that does NOT come down with the target. Do other proteins that make up the TOM complex also Co-IP with FLAG-GCase? It is critical to show at least one protein which does not bind. “Control” seems as untransfected cells (?) is not the only control needed for Co-IP experiments.

Fig 3B – is there a reason why HSC70 is a doublet only in the FLAG-transfected cells?

257 “GCase regulates OXPHOS complex I assembly in HEK cells and iPSC-derived dopaminergic neurons” This is an overstatement.

Only two CI subunits are identified, out of 45. Is the MS data supportive of reduced levels of other complex I subunits? Or is it just NDUFA10 and NDUFA9?

Reduction in levels of a few CI proteins is not sufficient to state that GCase REGULATES assembly, it could as easily regulate degradation or stability, and not assembly. Identification of one assembly factor – out of 13 known to this date – is supportive but not sufficient to make this statement. Pulse-chase experiments could address this.

4D: BN-PAGE gel needs to be shown in full as Complex I is a part of multiple assemblies, i.e. respirasome, SC2 and SC2 as well as monomer see <https://pubmed.ncbi.nlm.nih.gov/26928661/>) or <https://www.nature.com/articles/s41467-020-15467-7>)

there is a clear reduction in the abundance of SDHA in the NPCs L444P/L444P-2A-GC1 and GBA1-KO lanes, this is not addressed.

4E – is the activity of Complex I the only one affected ? Authors should measure activity of complexes IV, V and II (especially since there is a reduction in SHDA levels in BN-PAGE), to show that the reduction in activity is specific to complex I.

General response to the reviewers

We thank the reviewers for their careful reading of the manuscript and thoughtful suggestions. Please find our point-by-point responses to the comments below.

New key data added in the revised version of the manuscript:

- More in-depth studies on endogenously expressed GCCase were performed to confirm GCCase mitochondrial localization. Endogenous GCCase coimmunoprecipitation was optimized, and we confirmed the interactions with mitochondrial proteins as well as known cytosolic interactors.
- To strengthen the relevance of our findings related to the physiological and pathological role of GCCase in mitochondria, additional control experiments were performed. These include the generation of i) stable T-Rex HEK cells with inducible expression of the lysosomal hydrolase hexaminidase B and ii) stable T-Rex HEK cells with inducible expression of dMTS-WT-GCCase.
- More in-depth characterization of mitochondrial complex I function and integrity in T-Rex HEK cells as well as in patient neuronal cells.
- More in-depth characterization of organoid differentiation and maturation.
- In accordance with the reviewer's request, scRNAseq experiments were performed in individual organoids. More in-depth analysis of scRNAseq data and comparison with published datasets has been performed.
- We performed a timeline characterization and quantification of the neurodegenerative phenotype in midbrain organoids.
- All minor concerns have now been addressed.

Reviewer #1 (Remarks to the Author)

The data describes a novel potential role and localization of GCCase in mitochondria, and suggest that GCCase mutations may directly affect mitochondria function. Overall the data are interesting and highly novel, providing potential explanations for how GCCase mutants may cause neuronal dysfunction. However, there are several issues that reduce enthusiasm for publication. Most of the data showing the localization of GCCase in mitochondrial rely on artificial overexpression models of tagged proteins in HEK cells. More in-depth studies should be done on the endogenously expressed protein to rule out the possibility that mito localization doesn't occur from dramatic overexpression. Critical controls were missing in may assays. For example specificity was not addressed, through expressing another tagged hydrolase at similar levels as GCCase. The mitochondrial functional studies are well performed overall, however there is no evidence to suggest that the effects occur directly from GCCase in mitochondria as opposed to indirect effects from lysosomal or mitophagy dysfunction. There are also some conceptual hurdles that should be addressed better. One is the fact that Gaucher disease is an established lysosomal but not mitochondrial disorder, and the study doesn't provide the percent of GCCase distribution within lysosomal and mitochondrial, or how this changes with mutations. The idea that mitochondria represent a quality control mechanism for mutant GCCase clearance is interesting, however if true, it should be very difficult to visualize mutant GCCase interacting with LONP1 without protease inhibitors, since the mutant would be rapidly degraded. Finally, the CDDO and PFF experiments do not add much to the study due to specificity issues and artificial nature of PFFs. If these issues could be addressed, the work would represent an important advance in the field.

We thank the reviewer for these excellent points and suggestions. We provide our point-by-point responses below.

Concerning the major issues, we agree that further characterization at the endogenous level was somehow missing in our initial submission. To improve the relevance of our findings, we implemented and validated endogenous GCCase coimmunoprecipitation using magnetic beads and a crosslinking procedure. We were able to improve the efficiency of protein immunoprecipitation and to validate the interactions of GCCase with known cytosolic interactors (i.e., LIMP2) as well as mitochondrial proteins. Furthermore, we performed additional validation experiments in patient neuronal cells.

We agree with the comments concerning the need for an additional tagged lysosomal hydrolase to exclude overexpression-related artifacts. To address this issue, we generated and characterized stable T-Rex HEK cells with inducible expression of the lysosomal hydrolase hexaminidase B (Q4).

Concerning the comment “there is no evidence to suggest that the effects occur directly from GCCase in mitochondria as opposed to indirect effects from lysosomal or mitophagy dysfunction.”, we agree that other GCCase-related mechanisms could have an impact on mitochondria, and we do not claim that the mechanisms described in this work are the sole pathways driving mitochondrial dysfunction. As we previously summarized, possible mechanisms include autophagy (mitophagy), ER stress, lipid disturbances, and alpha-synuclein aggregation¹. However, thanks to the Reviewer’s suggestion, we have performed additional experiments using a dMTS-GCCase stable cell line, which strengthen our previous data showing that mitochondrial GCCase directly modulates CI. Furthermore, we show that inhibiting lysosome function with bafilomycin (Q14) has no effect on CI integrity. In summary, while we agree that other cellular processes may influence mitochondrial function in *GBA1* mutants, our findings suggest that GCCase plays a direct role in CI function and integrity. We have now expanded on this aspect in the discussion.

Concerning the “percentage of GCCase distribution within lysosomal and mitochondrial, or how this changes with mutations”, we provide a quantification in our detailed response below (Q7).

Concerning the interaction of GCCase with LONP1, we now provide clear evidence of such interaction at the endogenous level.

We agree on the artificial nature of the PFF model. However, as described below (Q24, Q25), we believe that this model still provides a useful way to examine A-SYN-related mechanisms and, in combination with CDDO, helps dissect the role of the GCCase/LONP1 interaction. Furthermore, these data support our hypothesis related to an overload of the mitochondrial proteostasis system, which may be relevant in a disease context. We agree with the reviewer concerning the need for higher resolution images. We now provide higher quality images.

Concerning the exact mitochondrial localization of A-SYN PFFs, we agree and we apologize for the confusion. Given the low resolution of confocal microscopy, we are unable to define the exact localization of exogenous A-SYN PFFs. However, increasing evidence shows that A-SYN accumulation may occur at mitochondrial membranes^{2,3,4}. Indeed, a very recent work show that intracellular seeding events occur preferentially on membrane surfaces, especially at mitochondrial membranes⁴. We have rephrased the text and discussion accordingly. As mitochondrial proteostasis can influence the homeostasis of cytosolic aggregation-prone proteins⁵, one hypothesis would be that LONP1 regulates A-SYN pathology at mitochondrial

membranes. Even though further studies are needed to dissect the mechanisms linking LONP1/GCase and cytosolic proteostasis, we believe that these data may be highly relevant for PD-related pathology.

These are the major issues, and below are included some additional suggestions, which I hope the authors find constructive and help to improve the work.

Comments

1. Please provide citation for the statement” The p.E326K variant displays a limited impact on GCase activity and a lower penetrance compared to the severe mutation p.L444P” line 93.

Citations have now been provided, as follows: "The p.E326K variant displays a limited impact on GCase activity and a lower penetrance compared to the severe mutation p.L444P^{6,7}.

2. (a) Please indicate whether the GCase tag is on the N or C terminus of the protein and (b) if the artificial tagged construct influences the normal processing of the enzyme, including glycosylation, cleavage, and transport to the lysosome.

The tag was positioned at the N terminus, 3 amino acids after the cleavage site of the leader sequence. These 3 amino acids are repeated after the tag to ensure the *GBA1* sequence is intact. We now provide additional information in the Methods section. Furthermore, we provide additional data showing GCase lysosomal localization, assessed by immunostaining, and GCase activity in T-Rex HEK cells overexpressing GCase. GCase activity was significantly increased in T-Rex HEK cells overexpressing WT-GCase, supporting a correct trafficking of GCase to the lysosome. Furthermore, GCase protein levels and activity were significantly decreased in E326K- and L444P- compared to WT-GCase T-Rex HEK cells, and the degree of reduction was in accordance with the severity of the mutation. Finally, to validate our approach and the use of the artificial tagged construct, we initially assessed GCase protein levels and lysosomal GCase activity of overexpressed tagged and untagged WT-GCase.

No significant differences between WT (untagged) and V5-FLAG-tagged GCase were observed.

Figure 1B, C, Supplementary Figure 1A.

The blot in Fig 1B appears clean, however one would expect to see multiple bands corresponding to different glycosylated forms of GCase at steady state.

We agree that multiple bands corresponding to the different glycosylated forms of GCase at steady state are not visible. This is due to the exposure time of the blot and amount of protein that was loaded. Longer exposure times are necessary to simultaneously show control or uninduced lines and GCase overexpressing cell lines. Below, two representative Western blots with different exposure times showing a comparable GCase pattern in control and T-Rex HEK cells expressing WT, E326K, or L444P GCase.

3. I would suggest including the densitometry and activity quantification from S1A, B, into the main figure as these are important data. Baseline activity from empty vector or minus Dox condition. Does the expression of L444P increase the activity over the base line minus Dox?

Thank you for the suggestion. The densitometry and activity quantification have now been included in Figure 1. Furthermore, we normalized GCase activity to empty vector control. The expression of L444P did not significantly increase the activity over the base line.

Figure 1 C, D.

4. An important control would include the overexpression of another FLAG-tagged lysosomal hydrolase at the same dramatically high levels as FLAG-GCase, to determine the specificity of the mitochondrial localization.

Thank you for this excellent suggestion. We agree with the reviewer and we generated stable T-Rex HEK cells with inducible expression of the lysosomal hydrolase (FLAG-tagged) Hexosaminidase B. As shown in *Supplementary Figure 3D-E*, overexpression did not lead to Hexosaminidase B mitochondria (mis)-localization.

5. In figure S3, the staining appears to have worked well, however it is difficult to see the colocalization of red and green channels. They appear non-overlapping in most of the images.

Supplementary Figure 3A shows GCase localization inside mitochondria, as assessed by expansion microscopy; due to the expansion of the sample and the nanoscale resolution, as TOM20 is on the outer mitochondrial membrane, no colocalization is expected. In standard confocal imaging (please see #7), colocalization is observed. In Supplementary Figure 3C, colocalization between LONP1 and GCase is shown. Such an interaction, along with the western blot data shown in Figure 2A, support the localization of GCase within the mitochondrial matrix.

6. In Figure 2E, the dMTS-GCase is critical in supporting the idea that GCase can traffic to mitochondria, however this figure should also include evidence that the dMTS mutant is stable and expressed at similar levels as compared to the MTS-versions.

Thank you for this suggestion. We performed these experiments and we provide representative Western blot images showing that the levels of expression of dMTS- GCase is comparable with WT-GCase.

Figure 2F

7. Quantification of the split-GCase mitochondrial localization is required to determine if wt and mutant GCase differ. They look relatively equal however one would expect the mutants to be slightly lower owing to their instability.

We performed quantification of the split-GCase images. As correctly pointed out by the reviewer, no significant difference was observed between WT and mutant GCase. While these data point toward an increased mitochondrial localization of mutant GCase, we do not feel confident in giving conclusive percentages of GCase intracellular distribution based on split-GFP methods. However, the western blot image in Figure 2A also indicates an increase of intramitochondrial GCase in severe mutants. To address this question, we assessed the proportion of intramitochondrial GCase by quantifying the SRCC of endogenous GCase/TOM20 in confocal images. We found a higher amount of GCase in mitochondrial in L444P/L444P mutants compared to control neurons. It is possible that the increased mitochondrial localization and interaction of mutant GCase with HSP60 and LONP1 reflects a mitochondria-based quality control system for cytoplasmic proteins as shown in ^{8, 9}. As the interactome analysis did not reveal significant differences in the interaction of WT and mutant GCase with proteins involved in mitochondrial import, we are currently investigating the possible mechanisms regulating mitochondrial localization in WT- and mutant GCase.

Supplementary Figure 5F.

8. The figure legend of S4 or text should define what each line means – what is “GC”?

Thank you for the comment. We clarified it.

9. Showing that the endogenously expressed GCase localizes with mitochondria is an important part of the study. Figure S4D and E are therefore critical and I would suggest to include in the main figure, however the quality of the blots should be improved. Endogenous GCase is difficult to pull down with commercially available antibodies. It is difficult to imagine such an efficient pull-down of unstable L444P GCase (lane 1) from the barely detectable input shown. Full blot should be shown here. Perhaps blotting for a known GCase interactor, such as LIMP2 as a positive control, would help, or colocalization with another mitochondrial marker to validate the findings. Quantification or some indication of this result is reproducible is also required.

We agree that this an important part of the study and it is correct that the pull-down of endogenous GCase is quite challenging. To address this point, we improved endogenous GCase co-immunoprecipitation using magnetic beads and a crosslinking procedure. Furthermore, co-immunoprecipitation was validated in GBA1 KO cells as well as using an IgG

control. We were able to validate the interactions of GCCase with known cytosolic interactors (i.e., LIMP2) as well as selected mitochondrial proteins from the proteomic dataset.

Figure 3G, Supplementary Figure 4E, F.

10. Clarification of the endogenous co-IP methods are required in the methods section, including the amount of lysate used, define “PD” line 497, and the GCCase antibody used for endogenous pull-down.

The endogenous co-IP methods are now described in detail in the methods section. Briefly, for endogenous GCCase co-immunoprecipitation, we incubated 7 mg of protein each sample with 25 μ l of DSS crosslinked A/G magnetic beads (Thermo Fisher, 88802-3) and 10 μ g of GBA1 antibody (MaxPap polyclonal, Abnova, H00002629-D01).

11. The immunofluorescence data in Figure S4G requires improvement and controls. For example, testing the antibody in a GCCase knock-out is required. Please indicate what percentage of GCCase is mitochondrial vs. lysosomal by adding in a colocalization study with LAMP1/ lysotracker etc.

We provide higher quality ExM images of endogenous LONP1/GCCase and a control using GBA1 knockout cells. We also provide mitochondrial vs. lysosomal localization of GCCase.

Supplementary Fig. 5C-F

12. Line 267, it is not clear how the data indicate a role for GCCase in CI assembly, since CI assembly was not assessed in this assay.

Thank you for this comment. The assembly state of CI was assessed by BN-PAGE analyses in Figures 4C-F and Supplementary Figure 6B. After extensive BN-PAGE analyses, we were unable to identify the characteristic accumulation of CI assembly intermediates that commonly occur upon disturbance of CI assembly. Moreover, the increased binding of mutant GCCase to mitochondrial proteases involved in quality control together with the decreased levels of TIMMDC1 in the mutants lead us instead to deduce that CI stability (rather than assembly) is undermined. Therefore, we have refined the role of GCCase throughout the manuscript text as a protein that regulates the maintenance of CI integrity and function.

13. Figure 4B, there appears to be some fragment of a blot image over top of the green plot in the graph.

Thank you for noticing it, we corrected it.

14. Figure 4D requires more replicates and statistical analysis. Here, it would be critical to include a control such as general lysosomal inhibitor, to determine if the effect is specific for GCCase path.

We agree with the reviewer and we performed additional experiments in T-Rex HEK cell lines as well as human NPCs and neurons with statistical analysis (Figure 4 C-F). An experiment with a bafilomycin control has also been performed. iPSC-derived neurons were treated with 10nM Bafilomycin for 6 h. Under these conditions, bafilomycin did not induce mitochondrial swelling neither significant changes in mitochondrial membrane potential nor respiration. We did not observe any impact of bafilomycin on CI integrity, as assessed by BN gels and IGA.

However, an impact of bafilomycin (and other lysosomal inhibitors) on mitochondria has been reported ^{10, 11, 12, 13}. This is somehow expected given the mechanical and functional link between lysosomes and mitochondria. These data suggest that lysosomal inhibitors have a number of off-target effects on mitochondria, making it difficult to dissect which are a direct consequence of inhibiting lysosomal function. Hence, we do not feel confident that using a lysosomal inhibitor could definitely rule out a possible link between lysosomes/any lysosomal enzyme and mitochondrial/CI function. On the other hand, while we report a novel mitochondrial link and function for GCCase, we do not claim that this link is unique to GCCase.

15. The decrease in mitochondrial function by GCCase mutants is interesting, however mitochondrial deficits have been previously demonstrated by other groups through other mechanisms. For example, it isn't clear how the authors can conclude

that this effect doesn't occur from a decline in mitophagy. Perhaps comparing to the dMTS-GCase mutant would help here.

Concerning the use of a dMTS-GCase mutant, thank you for the suggestion. To this end, we generated stable T-Rex HEK cells with inducible expression of dMTS-WT-GCase. As shown in *Supplementary Figure 7B*, dMTS-WT-GCase overexpression induces a decrease in CI activity compared to WT-GCase in the absence of changes in CIV activity. These results support the mitochondrial role of GCase in the CI integrity and function. As stated in the introduction, "impairment of mitochondrial metabolism and dynamics has been observed *in vivo* as well as in a variety of cellular models of GCase deficiency and patient neurons^{14, 15, 16, 17}". We do agree that other GBA1-related mechanisms could have an impact on mitochondria. Possible mechanisms include autophagy, ER stress, lipid disturbances, and alpha-synuclein aggregation. We have now expanded on this aspect in the discussion.

16. In Figure 5C, D, is there a feasible way to quantify the organoid diameter, so that we can get an idea of the distribution of sizes in the culture well? Size variability might play an important role in functional phenotypes, eg due to dead centers that are deprived of media.

As suggested, we measured the average organoid diameter, which is now shown in *Supplementary Figure 8B-C*.

17. Overall, the baseline characterization of the organoids appears very good, however better quantifications of TH+ and/or MAP2+ should be done. Figure S6A-E, and G all require quantification. Figure 5B-D, and F requires quantification.

We agree and we now provide quantification of TH+ neurons as well as A-SYN levels.

Figure 5F, G; Supplementary Figure 9C

18. In figure S6C, D, more studies and quantification are required to confirm that these structures are actually neuromelanin occurring from TH+ midbrain neurons.

We now provide DAB immunostaining combined with Fontana Masson showing TH+/neuromelanin+ neurons.

Supplementary Figure 8F

19. In organoid studies of Figure 5 and S6, there should be some indication that the data is reproducible between distinct organoid batches, not only between individual organoids. If different batches were used, this should be indicated in the figure legends.

The reviewer makes an important point. We now provide information concerning batches and individual organoids used in different experiments.

20. There are panel label errors in Figure 5.

Thank you for noticing it, we corrected the labeling.

21. Measurements of neurodegeneration should be done in the organoids at DIV42, to determine if the readouts at this time point are confounded by non-specific cell death processes.

This is an important point. We now provide a timeline quantification of TH+ neurons as well as A-SYN characterization in midbrain organoids.

Figure 5 F-H, Supplementary Figure 8C

22. In Figure S8C, it is difficult to see ‘high levels of GAD1 and TH’ line 343. Perhaps a quantitative assay such as ELISA or western blot would help here.

We apologies for the confusion; we identified one cellular cluster with high expression of GAD. Nonetheless, we performed western blots that show GAD1 levels in individual organoid lysates (figure below). Furthermore, in *Supplementary Figure 9C* we provide TH western blot in individual midbrain organoids generated from mutant and isogenic control iPSC lines.

GAD1 Rabbit, 1:1000 Proteintech, 10408-1-AP, RRID:AB_2107733

23. Figure S10- please indicate which genes are measured here in the figure legend.

Thank you for noticing it. Gene names are now included in the figure.

24. CDDO inhibits multiple pathways and the studies here do not add much to the overall study. It is also not clear why PFFs are needed since they induce artificial aggregation through mechanisms that are distinct from GBA1 mutations. These studies introduce some confusion to the work.

We employed 2-cyano-3,12-dioxooleana-1,9-dien-28-oic acid (CDDO-Me) to dissect the role of GCase/LONP1 interaction. CDDO-Me inhibits LONP1 protease¹⁸, but not the 26S proteasome¹⁹. Furthermore, CDDO-Me specifically inhibits LONP1's ATP-dependent proteolytic activity¹⁹. Hence, we believe this is a reliable model to dissect LONP1 function. Our hypothesis is that, in the presence of *GBA1* mutations, the increased proteolytic LONP1 activity promotes the degradation of misfolded/damaged intramitochondrial GCase. However, the burden of misfolded GCase may overload the capacity of mitochondrial proteases. We are currently working on the impact of intramitochondrial mutant GCase on mitochondrial proteostasis. Interestingly, our preliminary results show that CDDO induces an accumulation of insoluble proteins in WT-mitochondria similar to that occurring in mutant-mitochondria. We are pursuing these directions in detail for a follow up manuscript.

25. Ultrastructural studies are required to confirm that PFFs in figure 7A are intramitochondrial. Better resolution fluorescent images are also required, as it appears that large red colored puncta overwhelm the cytoplasm, which may produce artificial colocalization.

While we agree that exogenous A-SYN PFFs represent an artificial model, we believe that this model still provides a useful way to dissect A-SYN-related mechanisms. Furthermore, these data support our hypothesis related to an overload of the mitochondrial proteostasis system, which may be relevant in a disease context. We agree with the reviewer concerning the need for higher resolution images. We now provide higher quality images. Concerning the exact localization of A-SYN, we also agree that ultrastructural studies are required to confirm that PFFs are intramitochondrial. However, given the numerous experimental conditions, it would not be feasible to proceed with quantifications at the ultrastructural level. Furthermore, in light of a recent work showing that intracellular seeding events occur preferentially on membrane surfaces⁴, especially at mitochondrial membranes, we believe that these data may be highly

relevant for PD-related pathology. We have rephrased the results and discussion accordingly, and we described the limitation of standard imaging in assessing intracellular mitochondrial localization.

Reviewer #2 (Remarks to the Author)

In the manuscript “Glucocerebrosidase, a Parkinson’s disease-associated protein, is imported into mitochondria and regulates complex I assembly and function”, the authors aim to understand the role of Glucocerebrosidase (GCase) in mitochondria and its involvement in cellular energy metabolism through maintenance of respiratory chain complex I (CI) stability and integrity. The authors used wild-type and mutant GCase HEK 293 stable cell lines to show, mostly via a proteomic approach, the association between GCase and mitochondria and the interaction between mutant GCase and mitochondrial quality control proteins. They also used midbrain organoids differentiated from iPSCs carrying GBA1 mutations and an isogenic control to investigate the role of GCase in more relevant physiological conditions. The authors show the selective dysregulation of neuronal mitochondrial complex I (CI) in mutant DA neurons and speculate that this molecular dysfunction may be a key pathogenetic driver of Parkinson’s disease.

The major concepts of the manuscript are novel and could contribute to providing new insights into neurodegenerative diseases, and particularly into PD pathogenesis. The manuscript is well written and, overall, the conclusions are well supported. The efforts of the authors to study GCase in human neurons during differentiation from iPSCs is certainly commendable. However, a large part of the experiments are performed in HEK lines by protein overexpression. To make the manuscript suitable for publication in a top-tier journal such as Nature Communications, the authors should provide additional experiments to consolidate their findings regarding the role of GCase in human iPSC-derived neurons.

Major comments:

- 1. The manuscript relies on single cell data to obtain insights into cell-type specific mechanisms, but crucial details of the methods for these analyses are missing. For example, what cutoffs did the authors use to define a DE gene? Was the Harmony integration run based on time points or cell lines used? How did the data look before integration? How was the resolution established when defining clusters? How was the pseudotime analysis performed? How can claims about trajectories be made? Did the**

authors detect any cluster(s) that might indicate cell doublets? Violin or box plots illustrating the distribution of the number of expressed genes, UMI, and mitochondrial percentages should be shown.

The reviewer makes important points, and we addressed these comments:

- We included the cutoff for MAST analysis ($p_{adj} > 0,05$, $\log_{FC} > 2$).
- We clarified the integration method: we used the `IntegrateData` function from the Seurat package to reduce batch effects among cell lines at DIV 42 and among timepoints followed by Monocle.3 analysis.
- We now include data visualization before integration in *Supplementary Figure 10E*.
- Cells were clustered using a K-nearest neighbor (KNN) graph with a clustering resolution of 0.6, resulting in 21 clusters. Clusters were manually assigned and merged for visualization and analysis.
- We provide a detailed description of the pseudotime analysis and trajectories: "Pseudotime analysis was performed in the L444P/L444P 2 gene corrected line using Monocle 3 (version 1.0.0). Single cells isolated from 5 pooled organoids per time point were used for the analysis (day 42: 9692 cells, day 65: 8998 cells, day 100: 7422 cells). After data normalization and integration to reduce batch effects, the Monocle3 object was generated using the `as.cell_data_set` function, and cells were clustered using the `cluster_cells` function with a resolution of $1e-3$. Clusters were manually assigned, and claims about the states of the cells were made based on La Manno et al.²⁰.
- Concerning doublets, no cluster was found that could indicate cell doublets (`doubletFinder_v3` function):

- Violin and/or box plots illustrating the distribution of the number of expressed genes, UMI, and mitochondrial percentages are now included in *Supplementary Figure 10D*.

2. It is unclear from the figures and figure legends how many experimental batches or organoids were used. The statistical basis is also unclear in this manuscript: the authors often state “from three independent experiments”, but it is not obvious what this means. Do the authors present data from three separate organoids? Or did they pool data from several organoids generated on separate occasions? If the latter, from how many organoids?

The reviewer makes an important point, and we appreciate the confusion here. This point has now been clarified in the text more carefully. For single cell experiments at DIV 42, 5 pooled organoids from one batch per line were analyzed. For trajectory analysis, we pooled 5 organoids (L444P/L444P-2 GC) per time point from the same batch. We performed additional experiments where we analyzed individual organoids at DIV 42 from different batches, showing data reproducibility and minimal batch-to-batch variation.

Supplementary Figure 10 A-C

How many organoids gave rise to the findings shown in the main scRNA-seq figures? Data should not be reported in aggregate but in terms of individual organoids in order to experimentally substantiate the claims made in this study. The authors should also report the number of cells analyzed in the Results section.

We agree with the Reviewer, and we have performed additional scRNAseq experiments on individual organoids from a different batch. The non-integrated data UMAP visualization plot shows a similar cellular distribution (including specific markers of radial glia, neuronal precursors, neurons and dopaminergic neurons). Moreover, main cellular clusters were conserved among individual organoids. These data support the rationale of cluster aggregation for visualization and DE analysis. We now include the number of cells analyzed in the corresponding figure legends.

3. TH expression at single cell resolution is mainly detected at day 42 DIV (as shown in Fig. 6 D, Supplementary Fig. 8C). What happens to TH at later time points?

This is an important point. We now provide a timeline quantification of TH+ neurons as well as A-SYN quantification.

Figure 5E,F, Supplementary Figure 9C

4. It is unclear why the authors adapted the 2D differentiation protocol from Kim TW et al to generate a new and as yet unvalidated 3D midbrain differentiation protocol. It's not trivial develop a 3D protocol also starting from 2D, considering the high number of patterning factors used and the complexity of driving DA neuron differentiation. The authors should better characterize the generated midbrain organoids by showing the expression of postmitotic dopaminergic markers (e.g., Calb/GIRK2/DAT/ALDH1a) at protein level.

We agree with the Reviewer that implementing a new (3D) iPSC differentiation protocol is not trivial. As we properly refer to the original protocol stating that "Midbrain-like organoids were generated by a protocol adapted from Jo et al.", we have rephrased our statement "To improve midbrain organoid differentiation, we combined a previous method that promotes midbrain floor-plate formation ²¹ with a recent strategy based on the biphasic activation of WNT signaling, which promotes the reproducible 2D differentiation of midbrain DA neurons from pluripotent stem cells ²²". As suggested by the Reviewer, we have improved the basic characterization of midbrain organoid differentiation showing proper midbrain DA neuron emergence and maturation that was achieved using this modified protocol. The data are provided in *Supplementary Figure 8*.

Also, how does this protocol compare to other midbrain organoid protocols? The authors should project their single cell datasets onto available human midbrain organoid datasets, e.g., the recently published single cell dataset generated from human midbrain organoids by Fiorenzano et al (Nat Comm 2021).

We agree with this excellent suggestion. To assess the similarity between midbrain organoid generation protocols, we employed a published scRNAseq dataset by Fiorenzano et al. as a reference. Due to differences in the timepoints, we were unable to perform a timeline comparison between our dataset and the dataset obtained from silk-organoids, which show an increased homogeneity. Our analysis revealed a high similarity between floor plate markers along the radial glia clusters and a strong correlation of neuronal markers and dopaminergic markers. The cell type distribution shows the presence of radial glia, neuronal precursors, neurons and small clusters positive for astrocyte and oligodendrocyte markers in both datasets (*Supplementary Figure 11F*). Interestingly, while Fiorenzano et al organoids show a higher expression and persistence of radial glial and floor plate markers, cell distribution comparison showed a higher proportion of neurons in our dataset at 42 DIV. These data are now shown in *Supplementary Figure 11*.

5. Interestingly, the authors observe A-SYN accumulation at day 45 when DA neurons are not yet terminally differentiated. It would be beneficial for the manuscript to include an A-SYN immunofluorescence analysis with relative quantification at a later time point during midbrain organoid differentiation. Additionally, the immunofluorescence presented in Fig. 5G does not seem to reflect the total A-SYN levels assessed by ELISA. The authors should comment on this point.

Thank you for the interesting comments and suggestions. In contrast to other synaptic markers, we observed an early emergence of A-SYN+ neurons in developing organoids. Interestingly, pathological forms of A-SYN, namely, pSer-129 A-SYN, only appeared at later time points. To address this comment, we now provide tile images of A-SYN immunofluorescence at DIV 42, 65, and 100. These new images, with the corresponding TH staining, provide a better overview of whole organoids and confirm our previous results showing an increase in A-SYN levels in the mutant line over time, as well as ELISA-based assay. As requested, we attempted at measuring A-SYN immunofluorescence intensity levels/TH+ neurons. However, measuring intensity levels turned out to be quite unreliable. Hence, ELISA measurements are better suited for quantitative analysis of A-SYN in whole organoids. Furthermore, we performed pSer-129 A-SYN characterization that show increased p-A-SYN-Ser129+ neurons in GBA1 mutant organoids at 100 DIV. Finally, we provide western blot analysis of A-SYN soluble and insoluble fractions in long-term culture midbrain organoids, showing an increase of both the soluble and insoluble fractions in *GBA1* mutant organoids. Taken together, these data support the time-dependent increase in A-SYN levels in *GBA1* mutant organoids.

Figure 5E- H; Supplementary Figure 9C.

6. Accumulation of phosphorylated A-SYN is mainly observed in neurons in the form of Lewy bodies. Did the authors try to analyze phosphorylated A-SYN at day 45? I would strongly suggest that the authors analyze the accumulation of A-SYN and its phosphorylated form at later stages of differentiation, when neuromelanin is released and mature DA neurons are better modeled in a dish.

Thank you for the excellent comment and suggestion. Interestingly, pathological forms of A-SYN, namely, pSer-129 A-SYN, only appeared at later time points (Figure 5H). We now provide data relative to the pathological forms of A-SYN (Figure 5H, Supplementary Figure 9C).

Figure 5H, Supplementary Figure 9C.

Minor comments:

1. **Any residual floor plate (FP) progenitor cluster in Fig. 6A,B? Expression analysis at single cell level of the main markers of FP (FOXA2, LMX1A, LMX1B, CORIN, etc.) would be extremely useful.**

Thank you for the suggestion, we have now included these data in Supplementary Figure 10F.

2. **In light of changes in A-SYN levels, the authors should show the expression of SNCA in Fig. 6 D not in aggregate but by reporting isogenic control and mutant iPSC lines as individual samples.**

Thank you for the suggestion, we have now included these data in Supplementary Figure 12C.

3. **The authors cannot conclude from the analyses in Fig. 5E that “scRNAseq revealed a profound dysregulation of metabolic pathways” (page 10, lines 345–350). Gene ontology pathway analysis only reveals an enrichment in metabolic pathways relating to mitochondrial ATP-related functions. The authors should change the wording in both the Results and Discussion sections.**

Thank you for this comment, we agree and we have now rephrased the sentence: “scRNAseq reveals an enrichment in metabolic pathways...”

Reviewer #3 (Remarks to the Author):

In this manuscript, Baden et al, perform bulk proteomics, single-cell approaches, and IHC analysis in cell lines and iPSC cell and organoid models to investigate GCase-related neurodegeneration. The major findings are that a portion of GCase can enter mitochondria via the TOM import machinery. GCase promotes assembly of CI and interacts with HSP60 / LONP1. Notably, disease associated mutations appear to impair these functions. The manuscript does highlight a number of powerful modern techniques which are a strength. Overall, the Reviewer viewed this manuscript with some positively, but a few major and a collection of minor concerns need to be addressed before considering for publication. The clarity of the writing needs to also be improved.

Major concerns

1) If I am not mistaken, all biochemical experiments are based on overexpression of GCase and mutants, therefore it is a bit difficult to determine whether this is physiologically relevant or an artifact. I realize this is not an easy limitation to address.

This is an excellent and relevant comment. To improve the relevance of our findings, we implemented and validated endogenous GCase co-immunoprecipitation using magnetic beads and a crosslinking procedure. We were able to improve the efficiency of protein immunoprecipitation and to validate the interactions of GCase with known cytosolic interactors (i.e., LIMP2) as well as mitochondrial proteins. Furthermore, we have performed additional validation experiments in patient neuronal cells.

Figure 3G, Supplementary Figure 4E, F

2) Many of the experiments are either not controlled properly, or the controls need a better explanation in the text. i.e I am not sure what is the Ref library for all of the GO analyses, or what is the "CT" for Co-IP experiments. If CT is untransfected cell lysates, then that's a good control for the IP experiments but still ab it worrisome for the data presented in Figure 2.

The reviewer makes an important point, and we appreciate the confusion here. Concerning the GO analysis, we used Panther 14.0 analysis and Biological Processes as a GO aspect. For the interactome and proteomic analysis we filter out nonspecific proteins with a limma-based differential analysis comparing the conditions to empty vector controls. Statistically enriched hit and candidate proteins were analyzed using Panther pathway analysis software 14.0 (<http://www.pantherdb.org>; RRID:SCR_004869). The entire protein list was used as a background reference. For the transcriptomic analysis we clarified in the text: "For differentially expressed (DE) analysis, nonintegrated samples were analyzed using FindMarkers and MAST statistical tests ($p_{adj} > 0,05$, $\log_{FC} > 2$). DE genes were analyzed using Panther pathway analysis software 14.0, and Biological Processes as a GO aspect (<http://www.pantherdb.org>; RRID:SCR_004869)." Regarding Co-IP experiments, an empty vector control was employed (named CT). Selected candidates from the proteomics dataset have been also validated using untransfected cell lysates as well as at the endogenous level using *GBA1* KO cells. This has now been clarified in the text.

3) Every protein tested in the Co-IP experiments came down with the FLAG-tag.

We agree with the reviewer's comment, and we have now included Vinculin and TOM20 and as non-interacting protein control.

Supplementary Figure 3B.

Minor concerns

Are there blots showing endogenous GCCase in mitochondria? Fig 2A – weak band for “CT”.. ? that must be endogenous.

Yes, this is correct and we apology for the confusion. It is endogenous GCCase in T-REX HEK cells. We have now clarified this point. Furthermore, we performed additional experiments in human neuronal cells that show the presence of mitochondrial endogenous GCCase (*Supplementary Figure 4D*).

TMT analysis is not MS3-based and MS2 is just OK.

We were unable to find this issue.

Line 131 should be isobaric, not isotopic

Thank you for pointing this out, we corrected it.

1C – not sure what the control (for relative enrichment) is here. Input? WT pulldown (no flag expression)?

We apologies for the confusion, as stated above an empty vector control has been used.

1D – it isn't specified what is the reference of the GO analysis. All proteins in proteome? If so is that fair?

We agree with the reviewer's comment, and we have now included additional information. As both the proteomics and scRNAseq studies were unbiased and we did not employ a functionally-biased approach, such background list is appropriate. “Hit and candidate proteins were defined based on the following criteria. Candidate proteins were defined with an FDR ≤ 0.2 and an FC ≥ 1.5 , and hit proteins were based on an FDR < 0.05 and an FC > 2 .

Statistically enriched hit and candidate proteins were analyzed using Panther pathway analysis software 14.0. The entire human proteome was used as a background reference.”

Association of the GCCase with mitochondrial proteins is singled out while interactome with other organelles is ignored – other than previous literature is there a reason why lysosome or ER interactomes are ignored?

We agree with the reviewer that the most abundant GCCase interactors reside within other organelles, especially the ER and the Golgi apparatus. However, while GCCase interaction within such compartments is well established (these interactors were used as positive controls in our study), here we focused on the novel findings showing a potential GCCase mitochondrial localization (and function).

1E and S1E – again, not clear what is the control for this experiment. This data would be strengthened by an addition of negative i.e. protein that is not CO-IP’d with FLAG-tag, which would show specificity. It does look like EVERY protein tested comes down with the FLAG-tag, therefore showing a couple of proteins that do not is important.

As discussed in our previous point, T-Rex cells overexpressing the empty vector were used as a control. As suggested, Flag-IP has been validated using non-interacting protein controls.

Figure Supplementary 3B.

Fig 2C-E – clever very nice!

Thank you.

206: “GCCase is imported into mitochondria by the TOM protein import machinery”
• TOM import machinery consists of tom70, tom22, tom20, tom40, tom5, tom6 and tom7. HSP70 and Tim23 are not a part of the “TOM protein import machinery” - this needs to be cleared up the text.

Thank you for this excellent comment. We agree and we have now rephrased the statement.

• This figure also lacks controls i.e. Co-IP of a protein that does NOT come down with the target. Do other proteins that make up the TOM complex also Co-IP with FLAG-

GCCase? It is critical to show at least one protein which does not bind. “Control” seems as untransfected cells (?) is not the only control needed for Co-IP experiments.

We do agree and we have now included non-interacting controls.

Figure 3G, Supplementary Figure 4E

Fig 3B – is there a reason why HSC70 is a doublet only in the FLAG-transfected cells?

We also identified a band in the control, which, however, was less pronounced.

257 “GCCase regulates OXPHOS complex I assembly in HEK cells and iPSC-derived dopaminergic neurons” This is an overstatement. Only two CI subunits are identified, out of 45. Is the MS data supportive of reduced levels of other complex I subunits? Or is it just NDUFA10 and NDUFA9?

Thank you for this comment. According to the proteomic dataset, GCCase interacts with several CI subunits and assembly factors. The figure below provides a summary of all the CI subunits and assembly factors that were identified in the interactome study. Furthermore, we identified a decreased interaction of mutant GCCase with NDUFA9 and TIMMDC1.

Schematic representation of known CI subunits and assembly factors; in red all the proteins that were identified in the GCCase interactome.

Moreover, a proteome-wide analysis of the differentially expressed proteins (DEPs) from TMT-labeled WT-, E326K-, and L444P-GCCase T-Rex HEK cells (data shown in Supplementary Figure 2B) revealed increased abundance of CI subunits in the cells overexpressing WT- and mutant GCCase, indicating that CI levels are indeed modulated by GCCase. This is supported by more extensive SDS-PAGE and BN-PAGE analyses shown in Figure 4 and Supplementary Figure 6. Overall, these data indicate that GCCase is required for the structural and functional maintenance of mature CI and the respirasomes. We rephrased our statements in the text.

Reduction in levels of a few CI proteins is not sufficient to state that GCCase REGULATES assembly, it could as easily regulate degradation or stability, and not assembly. Identification of one assembly factor – out of 13 known to this date – is supportive but not sufficient to make this statement. Pulse-chase experiments could address this.

We thank the Reviewer for this thoughtful comment. In fact, after extensive BN-PAGE analyses we were unable to identify the characteristic accumulation of CI assembly intermediates that commonly occur upon disturbance of CI assembly. Moreover, the increased binding of mutant GCCase to mitochondrial proteases involved in quality control together with the decreased levels of TIMMDC1 in the mutants lead us instead to deduce that CI integrity is undermined. Therefore, we believe the Reviewer is right and we have lowered down our initial statement accordingly and included these new ideas in the discussion.

4D: BN-PAGE gel needs to be shown in full as Complex I is a part of multiple assemblies, i.e. respirasome, SC2 and SC2 as well as monomer see <https://pubmed.ncbi.nlm.nih.gov/26928661/> or <https://www.nature.com/articles/s41467-020-15467-7>) there is a clear reduction in the abundance of SDHA in the NPCs L444P/L444P-2A-GC1 and GBA1-KO lanes, this is not addressed.

The Reviewer is right, and we now show full BNE gels demonstrating that, in our experimental model in which mitochondrial samples were solubilized at a digitonin:protein ratio of 4:1 g/g, the practical totality of fully-assembled active CI is localized within supercomplexes (yet a negligible amount of free CI can be observed in NPCs).

Regarding the reduction in the abundance of SDHA (a CII subunit) in the NPCs L444P/L444P-2A-GC1 and GBA1-KO lanes, we did not address it because it was likely attributed to a loading error and this reduction was not reproduced between different experiments. In addition, CII activity was not affected in the mutant cells as shown now in *Supplementary Figure 7D* (see next response below). We have now included more a representative BN-PAGE analysis shown in *Supplementary Figure 6*, in which it can be seen that CII levels are unaffected.

4E – is the activity of Complex I the only one affected? Authors should measure activity of complexes IV, V and II (especially since there is a reduction in SHDA levels in BN-PAGE), to show that the reduction in activity is specific to complex I.

The Reviewer makes an important point. We do agree and we have not provided data relative to complexes II and IV in iPSC-derived neurons. Data are now shown in Supplementary Figure 7D.

References

1. Baden P, Yu C, Deleidi M. Insights into GBA Parkinson's disease pathology and therapy with induced pluripotent stem cell model systems. *Neurobiol Dis* **127**, 1-12 (2019).
2. Di Maio R, *et al.* alpha-Synuclein binds to TOM20 and inhibits mitochondrial protein import in Parkinson's disease. *Sci Transl Med* **8**, 342ra378 (2016).
3. Ryan T, *et al.* Cardiolipin exposure on the outer mitochondrial membrane modulates alpha-synuclein. *Nat Commun* **9**, 817 (2018).
4. Choi ML, *et al.* Pathological structural conversion of alpha-synuclein at the mitochondria induces neuronal toxicity. *Nat Neurosci* **25**, 1134-1148 (2022).
5. Lautenschlager J, *et al.* Intramitochondrial proteostasis is directly coupled to alpha-synuclein and amyloid beta1-42 pathologies. *J Biol Chem* **295**, 10138-10152 (2020).
6. Straniero L, *et al.* The SPID-GBA study: Sex distribution, Penetrance, Incidence, and Dementia in GBA-PD. *Neurol Genet* **6**, e523 (2020).
7. Montfort M, Chabas A, Vilageliu L, Grinberg D. Functional analysis of 13 GBA mutant alleles identified in Gaucher disease patients: Pathogenic changes and "modifier" polymorphisms. *Hum Mutat* **23**, 567-575 (2004).
8. Ruan L, *et al.* Cytosolic proteostasis through importing of misfolded proteins into mitochondria. *Nature* **543**, 443-446 (2017).
9. Li Y, *et al.* A mitochondrial FUNDC1/HSC70 interaction organizes the proteostatic stress response at the risk of cell morbidity. *EMBO J* **38**, (2019).

10. Dykstra KM, *et al.* Inhibiting autophagy targets human leukemic stem cells and hypoxic AML blasts by disrupting mitochondrial homeostasis. *Blood Adv* **5**, 2087-2100 (2021).
11. Teplova VV, Tonshin AA, Grigoriev PA, Saris NE, Salkinoja-Salonen MS. Bafilomycin A1 is a potassium ionophore that impairs mitochondrial functions. *J Bioenerg Biomembr* **39**, 321-329 (2007).
12. Redmann M, *et al.* Inhibition of autophagy with bafilomycin and chloroquine decreases mitochondrial quality and bioenergetic function in primary neurons. *Redox Biol* **11**, 73-81 (2017).
13. Zhdanov AV, Dmitriev RI, Papkovsky DB. Bafilomycin A1 activates respiration of neuronal cells via uncoupling associated with flickering depolarization of mitochondria. *Cell Mol Life Sci* **68**, 903-917 (2011).
14. Cleeter MW, *et al.* Glucocerebrosidase inhibition causes mitochondrial dysfunction and free radical damage. *Neurochem Int* **62**, 1-7 (2013).
15. Osellame LD, *et al.* Mitochondria and quality control defects in a mouse model of Gaucher disease--links to Parkinson's disease. *Cell metabolism* **17**, 941-953 (2013).
16. de la Mata M, *et al.* Pharmacological Chaperones and Coenzyme Q10 Treatment Improves Mutant beta-Glucocerebrosidase Activity and Mitochondrial Function in Neuronopathic Forms of Gaucher Disease. *Sci Rep* **5**, 10903 (2015).
17. Schondorf DC, *et al.* The NAD⁺ Precursor Nicotinamide Riboside Rescues Mitochondrial Defects and Neuronal Loss in iPSC and Fly Models of Parkinson's Disease. *Cell Rep* **23**, 2976-2988 (2018).
18. Gibellini L, *et al.* Inhibition of Lon protease by triterpenoids alters mitochondria and is associated to cell death in human cancer cells. *Oncotarget* **6**, 25466-25483 (2015).
19. Lee J, *et al.* Inhibition of mitochondrial LonP1 protease by allosteric blockade of ATP -binding and -hydrolysis via CDDO and its derivatives. *J Biol Chem*, 101719 (2022).
20. La Manno G, *et al.* Molecular Diversity of Midbrain Development in Mouse, Human, and Stem Cells. *Cell* **167**, 566-580 e519 (2016).
21. Jo J, *et al.* Midbrain-like Organoids from Human Pluripotent Stem Cells Contain Functional Dopaminergic and Neuromelanin-Producing Neurons. *Cell Stem Cell* **19**, 248-257 (2016).
22. Kim TW, *et al.* Biphase Activation of WNT Signaling Facilitates the Derivation of Midbrain Dopamine Neurons from hESCs for Translational Use. *Cell Stem Cell* **28**, 343-355 e345 (2021).

REVIEWER COMMENTS

Reviewer #1 (Remarks to the Author):

The authors have addressed all of my major concerns. The co-IP with endogenous GCCase (Fig 4G) has been improved with quantification shown. The characterization of the organoids is more complete. The addition of Hex B overexpressing lines is helpful. One suggestion would be to include a few more cells in this Hex B image however. While the image shown is clear, the entire cell body is not shown- it is cut off. It would be more informative / convincing to show the entire cell body and multiple cells in one field of view.

Reviewer #2 (Remarks to the Author):

Overall, the manuscript has gained clarity and focus, and it has considerably improved over the past iteration. The authors have adequately addressed of the concerns voiced in earlier review.

Reviewer #3 (Remarks to the Author):

The authors have adequately addressed some of my previous concerns. However, the presented data still fall short of strongly supporting their conclusions. For example, there is no strong data / evidence that GCCase localizes to mitochondria. It's a bit of an overstatement to claim that the functional data can simply be explained as being directly due to GCCase-related mitochondrial dysfunction.

My remaining specific concerns are as follows.

1. Fig.1F shows IP of FLAG-GCCase can isolate entire mitochondria (including both OMM, IMM and matrix). This is hard to understand. At the same time, Fig. 2B suggests GCCase is localized within the

matrix. If true, how can GCCase interact with all those OMM, IMM and matrix proteins? An additional control such as FLAG-GFP would increase rigor and represent a better negative control than empty vector.

2. Lots of key controls are missing in Fig. 2A. (1), the authors need to show the digitonin treatment worked, which need to show the levels of OMM, IMM and matrix proteins with and without digitonin treatment. (2) the authors need to show the protein levels in two more conditions: PK (+) digitonin (-) and PK (-) digitonin (+). (3) Also the same experiment with FLAG-GFP and FLAG-GCCase would be important to include.

3. I'm confused with the data shown in Fig. 2B. Do the authors suggest GCCase is a transmembrane protein?

4. Fig. 2C, D & E. These data are of rather low quality and fail to support mitochondrial localization of GCCase. Additional controls are needed. For example, MTS-GFP1-10 is very likely to bind GFP11-GCCase before it is trafficked to mitochondria. Thus, all the mitochondrial GCCase is just dragged by MTS-GFP1-10. At a minimum the authors should include GFP1-10 without any MTS.

5. Fig. 3A, I failed to find any GCCase in this cartoon and Fig. 3H, the pattern does not look like mitochondrial.

6. Unfortunately, the functional data cannot be simply explained as being solely related to GCCase in mitochondrial function.

Reviewer #1 (Remarks to the Author):

The authors have addressed all of my major concerns. The co-IP with endogenous GCCase (Fig 4G) has been improved with quantification shown. The characterization of the organoids is more complete. The addition of Hex B overexpressing lines is helpful. One suggestion would be to include a few more cells in this Hex B image however. While the image shown is clear, the entire cell body is not shown- it is cut off. It would be more informative / convincing to show the entire cell body and multiple cells in one field of view.

We thank the reviewer for the positive evaluation. We agree with the reviewer's comment and we now provide additional images of the HexB overexpressing lines.

Reviewer #2 (Remarks to the Author):

Overall, the manuscript has gained clarity and focus, and it has considerably improved over the past iteration. The authors have adequately addressed of the concerns voiced in earlier review.

We thank the reviewer for the positive evaluation.

Reviewer #3 (Remarks to the Author):

The authors have adequately addressed some of my previous concerns. However, the presented data still fall short of strongly supporting their conclusions. For example, there is no strong data / evidence that GCCase localizes to mitochondria. It's a bit of an overstatement to claim that the functional data can simply be explained as being directly due to GCCase-related mitochondrial dysfunction.

My remaining specific concerns are as follows.

1. Fig.1F shows IP of FLAG-GCCase can isolate entire mitochondria (including both OMM, IMM and matrix). This is hard to understand. At the same time, Fig. 2B suggests GCCase is localized within the matrix. If true, how can GCCase interact with all those OMM, IMM and matrix proteins? An additional control such as FLAG-GFP would increase rigor and represent a better negative control.

In Fig.2A we displayed a graphical representation of all proteins identified as GCCase interactors by TMT proteomics; interestingly, some of these proteins are involved in mitochondrial protein import and processing (cytosolic chaperones as well as mitochondrial import and matrix proteins). As GCCase has an iMTS-Is, the interaction with proteins in the OMM, IMM, and matrix is expected.

Concerning the suggestion to include an additional negative control, we apologize if this was unclear. In the latest version of the manuscript we already provided several control conditions. These include a Flag-V5 control line, a Flag-PD experiment against non-interacting proteins, an empty-vector control, and a reverse pull-down. Importantly, we provided an extensive validation of the interacting proteins at the endogenous level using multiple patient induced pluripotent stem cells. In these experiments, an IgG control and a *GBA1* knockout iPSC line were used as controls. Hence, we believe that the control conditions we employed throughout our work provide a strong evidence for the mitochondrial localization of GCCase.

2. Lots of key controls are missing in Fig. 2A. (1), the authors need to show the digitonin treatment worked, which need to show the levels of OMM, IMM and matrix proteins with and without digitonin treatment. (2) the authors need to show the protein levels in two more conditions: PK (+) digitonin (-) and PK (-) digitonin (+). (3) Also the same experiment with FLAG-GFP and FLAG-GCase would be important to include.

We thank the reviewer for this suggestion and we now provide a control experiment with digitonin and increasing concentrations of proteinase K showing OMM (Tom20), IMM (TIM23), and matrix proteins (LonP1) in the different fractions (Supplementary Figure 3A). Furthermore, in the latest version of the manuscript, we performed a subcellular fractionation experiment in isogenic iPSC-derived neural precursor cells (NPCs) generated from L444P/L444P *GBA1* iPSCs and isogenic controls, which confirmed the intramitochondrial localization of GCase (Supplementary Figure 4D). Hence, the control conditions we provided throughout our work strongly support the mitochondrial localization of GCase.

3. I'm confused with the data shown in Fig. 2B. Do the authors suggest GCase is a transmembrane protein?

We thank the reviewer for this comment. With this data, we do not suggest that GCase is a transmembrane protein. It is known that some matrix proteins contain iMTS-Is (PMID: 29382700, PMID: 20729931).

4. Fig. 2C, D & E. These data are of rather low quality and fail to support mitochondrial localization of GCase. Additional controls are needed. For example, MTS-GFP1-10 is very likely to bind GFP11-GCase before it is trafficked to mitochondria. Thus, all the mitochondrial GCase is just dragged by MTS-GFP1-10. At a minimum the authors should include GFP1-10 without any MTS.

We thank the reviewer for this comment. We acknowledge and apologize for the low quality of the image. We now included higher-quality images.

Concerning the question relative to additional controls, while we agree that a small fraction of the interaction of MTS-GFP1-10 with GFP11-GCase may occur outside mitochondria, it is unlikely to colocalize with MitoTracker Red, if it is not indeed mitochondria-specific. Furthermore, if MTS-GFP1-10 bound GFP11-GCase before trafficking to mitochondria, we would also expect that MTS-GFP1-10 binds dMTS-GCase and drag it inside mitochondria. We were not able to find any positive signal in the dMTS-GCase line, indicating that the protein is unable to bind MTS-GFP1-10 outside mitochondria (Figure 2E). Additionally, while we agree that cytosolic GFP1-10 could be used as a control to confirm the positive binding of GFP1-10 with 11-GFP, we already used MTS-GCase as a control construct. MTS-GCase, which contains the mitochondrial targeting sequence of Cox8A, binds MTS-GFP1-10 and colocalizes with MitoTracker Red.

5. Fig. 3A, I failed to find any GCase in this cartoon and Fig. 3H, the pattern does not look like mitochondrial.

We agree with the reviewer that the cartoon was somehow unclear. The cartoon is a simplified graphic representation of all proteins (cytosolic chaperones as well as mitochondrial import and matrix proteins) that have been identified as GCase interactors by TMT proteomics. We have now clarified this in the figure legend and modified the figure accordingly.

In Figure 3H, we performed expansion microscopy (ExM) to confirm the interaction of LonP1/GCase. After ExM, cells were imaged using confocal microscopy with a 100x objective, which gives a final expansion of 400x and an expected resolution of ~70 nm. Although the

loss of mitochondrial pattern is not an issue when visualizing mitochondrial membrane structures such as cristae and complexes, it is more challenging to visualize the specific mitochondrial pattern of matrix proteins without membrane reference markers in the staining. To confirm that the pattern of LONP1 is indeed mitochondrial, we initially performed a control staining of LONP1/TOM20 before and after ExM. While the digestion step needed for ExM decreases the signal of the antibody, the pattern of LONP1 remains mitochondrial.

Conventional confocal microscopy and expansion microscopy were performed in HEK cells following the protocols described in the methods section. TOM20, green; LONP1, red. Scale Bar, 10 μ m.

6. Unfortunately, the functional data cannot be simply explained as being solely related to GCCase in mitochondrial function.

This point has already been addressed in our latest version of the manuscript. We agree that additional *GBA1*-related mechanisms may have an impact on mitochondria, and we do not claim that the mechanisms described in this work are the sole pathways driving mitochondrial dysfunction in *GBA1*-PD. As we previously summarized, possible mechanisms include autophagy (mitophagy), ER stress, lipid disturbances, and alpha-synuclein aggregation (PMID: 30711484). However, our data clearly show that GCCase plays a direct role in mitochondrial function. In the latest revision of the manuscript, using a T-Rex HEK cell line overexpressing dMTS-GCCase, we confirmed that the increase in CI activity is directly linked to GCCase mitochondrial localization (Supplementary Figure 7). Furthermore, we provided evidence that WT GCCase overexpression leads to an increase in CI subunits compared to control lines and L444P GCCase (Figure 3). This is consistent with our findings showing increased complex I activity and oxygen consumption rate upon GCCase overexpression (Supplementary Figure 7). It is important to note that *GBA1* mutations are a genetic risk factor for the disease. While we agree that other cellular processes influence mitochondrial function in *GBA1* mutants, our findings suggest that GCCase plays a direct role in CI function and integrity. Several findings support a key role for CI in PD. Hence, CI dysfunction observed in *GBA1* carriers may underly such as increased disease risk in an aging individual. We already expanded on these aspects in the discussion.

REVIEWERS' COMMENTS

Reviewer #3 (Remarks to the Author):

The authors have addressed all my concerns, I support publication.